# Strong angular and spectral narrowing of electroluminescence in an integrated Tamm-plasmon-driven halide perovskite LED

Zher Ying Ooi[1], Alberto Jiménez-Solano [2,3], Krzysztof Gałkowski[4,5,6], Yuqi Sun [4], Jordi Ferrer Orri [4,7], Kyle Frohna [4], Hayden Salway[1], Simon Kahmann[1,4], Shenyu Nie[1], Guadalupe Vega [3,8], Shaoni Kar [4], Michał P. Nowak [9], Sebastian Maćkowski[5], Piotr Nyga [9], Caterina Ducati [7], Neil C. Greenham [4], Bettina V. Lotsch [2,10,11], Miguel Anaya [1,8] ✉ & Samuel D. Stranks [1,4] ✉

Next-generation light-emitting applications such as displays and optical communications require judicious control over emitted light, including intensity and angular dispersion. To date, this remains a challenge as conventional methods require cumbersome optics. Here, we report highly directional and enhanced electroluminescence from a solution-processed quasi-2-dimensional halide perovskite light-emitting diode by building a device architecture to exploit hybrid plasmonic-photonic Tamm plasmon modes. By exploiting the processing and bandgap tunability of the halide perovskite device layers, we construct the device stack to optimise both optical and charge-injection properties, leading to narrow forward electroluminescence with an angular full-width half-maximum of 36.6° compared with the conventional isotropic control device of 143.9°, and narrow electroluminescence spectral full-width half-maximum of 12.1 nm. The device design is versatile and tunable to work with emission lines covering the visible spectrum with desired directionality, thus providing a promising route to modular, inexpensive, and directional operating light-emitting devices.

In the past decade, light-emitting diodes (LEDs) have dominated the light emission market in applications such as lighting, indicators and displays thanks to their outstanding performance including long lifetimes, low energy consumption, fast switching, small size and high robustness[1,2]. Whereas crystalline epitaxially grown III-V

semiconductors[3,4] are still dominating the lighting market, new generation emitting materials such as solution-processed quantum dots[5], organics[6] and polymers[7] are now finding traction particularly in the display market. More recently, solution-processible halide perovskites have emerged as popular optoelectronic materials with remarkable

[1]Department of Chemical Engineering and Biotechnology, University of Cambridge, Cambridge, UK. [2]Max Planck Institute for Solid State Research, Heisenbergstrasse 1, 70569 Stuttgart, Germany. [3]Departamento de Física, Universidad de Córdoba, Edificio Einstein (C2), Campus de Rabanales, 14071 Córdoba, Spain. [4]Cavendish Laboratory, University of Cambridge, Cambridge, UK. [5]Institute of Physics, Faculty of Physics, Astronomy and Informatics, Nicolaus Copernicus University, Toruń, Poland. [6]Department of Experimental Physics, Faculty of Fundamental Problems of Technology, Wrocław University of Science and Technology, Wrocław, Poland. [7]Department of Materials Science and Metallurgy, University of Cambridge, Cambridge, UK. [8]Departamento Física de la Materia Condensada, Instituto de Ciencia de Materiales de Sevilla, Universidad de Sevilla–CSIC, Calle Américo Vespucio 49, Sevilla 41012, Spain. [9]Institute of Optoelectronics, Military University of Technology, Warsaw, Poland. [10]Department of Chemistry, Ludwig-Maximilians-Universität (LMU), Butenandtstrasse 5-13, 81377 Munich, Germany. [11]e-conversion, Lichtenbergstrasse 4a, 85748 Garching, Germany. ✉e-mail: ma811@cam.ac.uk; sds65@cam.ac.uk

properties, including bandgap tunability, high carrier mobility and high luminescence efficiency[8,9], which makes them promising candidates for the next generation lighting technologies. To fully exploit the properties of these materials in a range of light applications, absolute control of the emission properties via the implementation of refined light management methods will be crucial.

Conventional photon management strategies include implementation of plasmonic waveguides[10–16], photonic cavities[17–21], and hybrid cavities[22]. Plasmonic waveguides have strong light confinement for strong enhancement of optical field and optical force but the plasmonic metal attenuates electromagnetic fields thus forming short-ranged plasmonic effects[23,24]. Photonic systems have longer range, however, the confinement across the emitting layer is typically lower than the plasmonic analogues[22,25]. A long-ranged and strongly confined hybrid plasmonic-photonic structure is an attractive solution. Tamm plasmons are localised surface states that are confined at the metal (plasmonic)-photonic-crystal interface, with metal deposited directly on the high refractive index layer of the photonic crystal[26,27]. The Tamm plasmon resonance can be tuned across the photonic stopband of the photonic crystal by varying the thickness of the top high refractive index layer at the interface[27]. Unlike conventional surface plasmons, Tamm plasmons form both transverse electric and magnetic mode polarisations with dispersion within the light cone, and thus can be optically excited without additional optical prisms or gratings[27]. The Tamm plasmon modes are found in simple planar structures which are relatively easy to fabricate, easy to design and tune, and could easily transform into a device architecture. These remarkable properties of Tamm plasmon modes have prompted their development for a variety of optical applications such as optical coatings with dye-doped nanospheres[28], III-V semiconductor lasers[29], organic solar cells[30] and quantum dot-based single photon sources[31].

In this work, we employ a halide perovskite light-emitting diode (PeLED) device driven by Tamm plasmon modes for precise angular and colour control. With careful control of the structure, we tune the Tamm plasmon resonance wavelength to match the perovskite electroluminescence (EL) peak and thus confine the Tamm plasmon modes within the entire perovskite layer. The Tamm plasmon resonance modes enhance the perovskite EL in forward direction by a factor of 1.8 compared to the reference, and outcouple light modes efficiently with narrow and controllable angular dispersion with angular full-width half-maximum (FWHM) of 36.6° compared to 143.9° for the control. As a plasmonic system, the Tamm plasmon modes show stronger confinement of the electromagnetic field than the full photonic microcavities, thus attractive for PeLEDs as the perovskite film is typically thin. The approach is versatile and tuneable across different emitting energies and angles, thus opening avenues for wider applications in display and light sources which require fine control of the angular dispersion and intensity of emitted light.

## Results

### Development of perovskite-based Tamm plasmon structure

We first employ a transfer matrix model powered by a genetic algorithm (see Methods for further details) to establish the optimum combination of materials and thickness of each layer to achieve a strong Tamm plasmon resonance confined across the entire quasi-2-dimensional (2D) perovskite layer (Supplementary Fig. 1). A quasi-2D $(PEA)_2(CsPbBr_3)_{n-1}PbBr_4$ perovskite (PEA: 2-phenylethylamine, Cs: caesium, Pb: lead, Br: bromide) is employed and optimised to realise a smooth thin film with photoluminescence (PL) peak wavelength around 510 nm (Supplementary Fig. 1). This material offers high brightness, photoluminescence quantum yield and external quantum efficiency (EQE) when incorporated in LEDs due to dielectric and quantum confinement effects governed by the formation of lower dimensional halide perovskite structures[32,33] (Supplementary Fig. 2). We show using simulations that using silver (Ag) as the Tamm plasmon metal layer

achieves the highest enhancement of electric field within the perovskite layer compared with aluminium (Al) and gold (Au) (Supplementary Fig. 3). This is due to small ohmic loss of Ag (low imaginary permittivity or dielectric constant) across the visible spectrum[34].

According to the optimised simulation design, we then experimentally fabricated the perovskite-based Tamm-plasmon structure. As shown in the cross-sectional high-angle annular dark field scanning transmission electron microscopy (HAADF-STEM) image (Fig. 1a) and energy dispersive X-ray (EDX) spectroscopy chemical composition map (Fig. 1b), the optimised Tamm-plasmon-based perovskite structure has smooth and uniform interfaces across each layer and the thickness of each layer closely matches the optimised design from simulation (Fig. 1c). The perovskite-based Tamm-plasmon structure employed here is made up of a 1-dimensional photonic crystal comprising 3 pairs of alternating titanium dioxide ($TiO_2$) and silicon dioxide ($SiO_2$) layers, a thin (roughly 10 nm to minimise optical effects) layer of polyvinylcarbazole (PVK), a quasi-2D perovskite film and a silver layer. The thin PVK layer is added to ensure consistency of perovskite growth on different substrates and will act as the eventual hole transport layer within the PeLEDs. The quasi-2D perovskite film with a refractive index of $n = 2.05$ (at 510 nm) replaces the $TiO_2$ layer ($n = 2.35$ at 510 nm) as the bottom photonic crystal layer at the metal-photonic-crystal interface (see Supplementary Note 1 for details of photonic crystal). The quasi-2D perovskite film is placed at the metal-photonic-crystal interface as the Tamm plasmon modes are strongly confined at the metal-photonic-crystal interface. From the simulation shown in Fig. 1c, we observe that the Tamm plasmon resonance enhances and confines electric field intensity at the metal-photonic-crystal interface, and the enhancement spans the entire perovskite layer when considering the emission wavelength of 510 nm.

Fine tuning of the Tamm plasmon resonance is realised by varying the thickness of the final high refractive index layer at the metal-photonic-crystal interface[26,27,34]. We finely manipulate the perovskite thicknesses between 20 nm to 54 nm by tuning the concentration of perovskite precursors (Fig. 1d). The resulting perovskite thicknesses are estimated with atomic force microscopy (AFM) (Supplementary Fig. 4). The AFM thickness measurements are in excellent agreement with the HAADF-STEM imaging and the fitted simulation results from the Tamm plasmon resonance wavelength, with resulting error within ±3 nm (Fig. 1e). We note that the Tamm plasmon resonance wavelengths from Fig. 1d, e are measured using an integrating sphere where the sample is tilted at 8° (see Methods for details); the Tamm plasmon resonance at 0° is estimated to be ca. 3 nm red-shifted from this value. In Fig. 1e, it is seen that the experimentally measured Tamm plasmon resonance linearly red shifts from 485 nm to 565 nm as the perovskite thickness increases from 20 nm to 54 nm. From these samples, we found that a perovskite thickness of 26 nm in the perovskite-based Tamm plasmon stack gives a matching Tamm plasmon resonance with the perovskite film PL wavelength (dashed black line).

### Directionality of the perovskite-based Tamm plasmon structure

To contextualise the properties of our perovskite-based Tamm plasmon structure, we first define a reference perovskite structure as the glass/PVK/quasi-2D perovskite/Ag stack which has identical layers to the Tamm-plasmon structure discussed above but eliminating the $TiO_2/SiO_2$ layers that compromise the photonic crystal. We observe a shortened PL lifetime across a range of fluences in the reference-perovskite stack (lifetime of 1.5 ns) compared to the reference-perovskite structure without Ag (lifetime of 3.7 ns) (Supplementary Fig. 5), indicating the direct deposition of Ag on perovskite quenches the PL through increased non-radiative recombination at the interface[35,36]. By the reference perovskite definition, we rule out the effect of the Tamm plasmon resonance in our reference-perovskite stack while retaining the quenching effect of Ag on the perovskite. As shown in Fig. 2a, b, the PL of our reference-perovskite structure follows

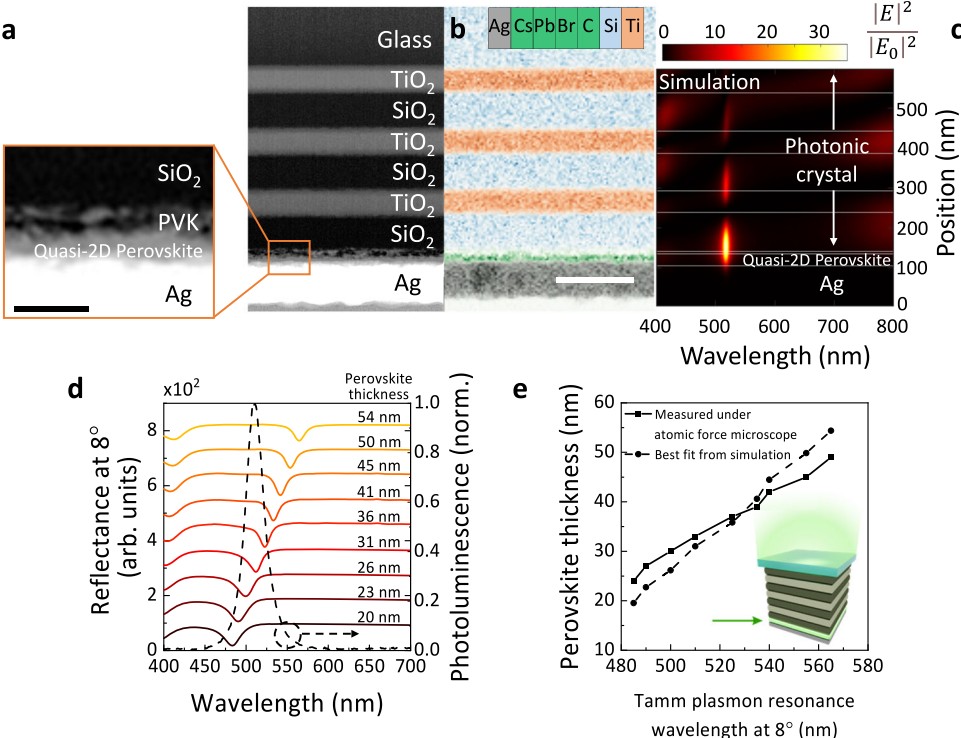

**Fig. 1 | Development of the perovskite-based Tamm plasmon structure. a** Cross-sectional HAADF-STEM image of the perovskite-based Tamm plasmon structure. Left: Magnified image of the metal-quasi-2D perovskite interface. The scale bar shows 50 nm. A similar structure is fabricated as the reference-perovskite structure (without TiO₂/SiO₂ layer pairs) to ensure results are not affected by unwanted effects, e.g., PL quenching due to metal deposition. The thickness of each layer is TiO₂ = (70 ± 2) nm, SiO₂ = (90 ± 2) nm, quasi-2D perovskite + PVK = (34 ± 5) nm, Ag = (100 ± 5) nm. **b** EDX chemical map of perovskite-based Tamm plasmon structure. Areas rich in Ag, Cs, Pb, Br, C, Si and Ti are marked in the legend. The scale bar shows 200 nm. The observed halo features at the titanium (Ti)-rich interface are found to be a slight compositional gradient of Ti, while the oxygen composition remains constant at the interface. **c** Simulated electric field enhancement, $\frac{|E|^2}{|E_0|^2}$ of the narrow-angle-Tamm-plasmon-perovskite structure. **d** Experimental optimisation of perovskite-based Tamm-plasmon structure by varying the perovskite thicknesses between 20 nm to 54 nm (Tamm plasmon resonance wavelength best fitted with transfer matrix model to estimate thickness of perovskite). Tamm plasmon resonance dip acquired from reflectance measurements at 8°. PL of quasi-2D perovskite thin film on glass is shown as dashed line. **e** Tamm plasmon resonance wavelength at 8° versus perovskite film thickness. Perovskite thickness is measured under AFM (solid line) and estimated from Tamm plasmon resonance wavelength best fitted with simulation (dashed line). Inset: schematic of the perovskite-based Tamm-plasmon structure with a green arrow indicating the quasi-2D perovskite layer.

a Lambertian emission profile, which is comparable to the finite-difference time-domain simulation (Fig. 2c) and is typical for isotropic emitters[37]. The PL maximum of the reference-perovskite structure is constant at 510 nm across all angles.

Comparing Fig. 2a, d, the PL linewidth of the perovskite-based Tamm plasmon structure with perovskite thickness of 26 nm (FWHM of 16.9 nm) is significantly narrower than the reference-perovskite sample (FWHM of 29.1 nm). We observe that the PL intensity of the perovskite-based Tamm plasmon structure with perovskite thickness of 26 nm is enhanced and peaks at 510 nm in the forward direction (0° to normal) (Fig. 2d). Thus, we refer to this sample as the narrow-angle-Tamm-plasmon-perovskite stack because the Tamm plasmon resonance matches the perovskite PL at narrow angles. As the angle increases, the PL intensity is suppressed and slightly blue-shifted. This observation is concomitant with the blue-shifting of the Tamm plasmon resonance at increasing angles in a Tamm plasmon system[27]. As shown in Fig. 2e, the narrow-angle-Tamm-plasmon-perovskite exhibits sharp emission directionality with angular FWHM of 44.7° compared to 124.5° for the reference perovskite with the same excitation power density (excited with a 405 nm laser). While all samples were excited at the same excitation intensity, the narrow-angle-Tamm-plasmon-perovskite shows strong enhancement by a factor of 2.6 in forward emission within a solid angle of ±15° perpendicular to the sample surface (calculations in Methods), which is crucial in applications that require forward emission.

In Supplementary Fig. 5, we observe no significant changes in PL lifetime of the narrow-angle-Tamm-plasmon-perovskite compared with the reference perovskite structure considering small sample-to-sample variation, which is consistent with other planar confined systems operating under weak confinement regime and reported very small <10% changes[38–41]. The simulated absorption across the entire perovskite layer at the excitation wavelength of 405 nm within both the reference-perovskite structure and the narrow-angle-Tamm-plasmon-perovskite structure are comparable (Supplementary Fig. 6), thus showing the forward enhancement in PL is primarily due to the Tamm plasmon modes resonating at the emission wavelength of 510 nm rather than photon recycling by re-absorption-re-emission events. We also note that both samples have uniform spatial uniformity of PL (Supplementary Fig. 7). Indeed, the enhanced directional emission of the narrow-angle-Tamm-plasmon-perovskite structure observed experimentally is also comparable to the simulation shown in Fig. 2f. Thus, we report strongly directional emission with our narrow-angle-Tamm-plasmon-perovskite structure.

We demonstrate that the perovskite-based Tamm plasmon structure offers versatile adjustment of the emission directionality between 0° to 40° by tuning the quasi-2D perovskite thickness between 26 nm and 54 nm (Supplementary Fig. 8). Taking the perovskite-based Tamm plasmon sample with perovskite thickness of 50 nm as an example, we observe a significantly red-shifted PL at small angles (Fig. 2g), which matches the Tamm plasmon resonance mode

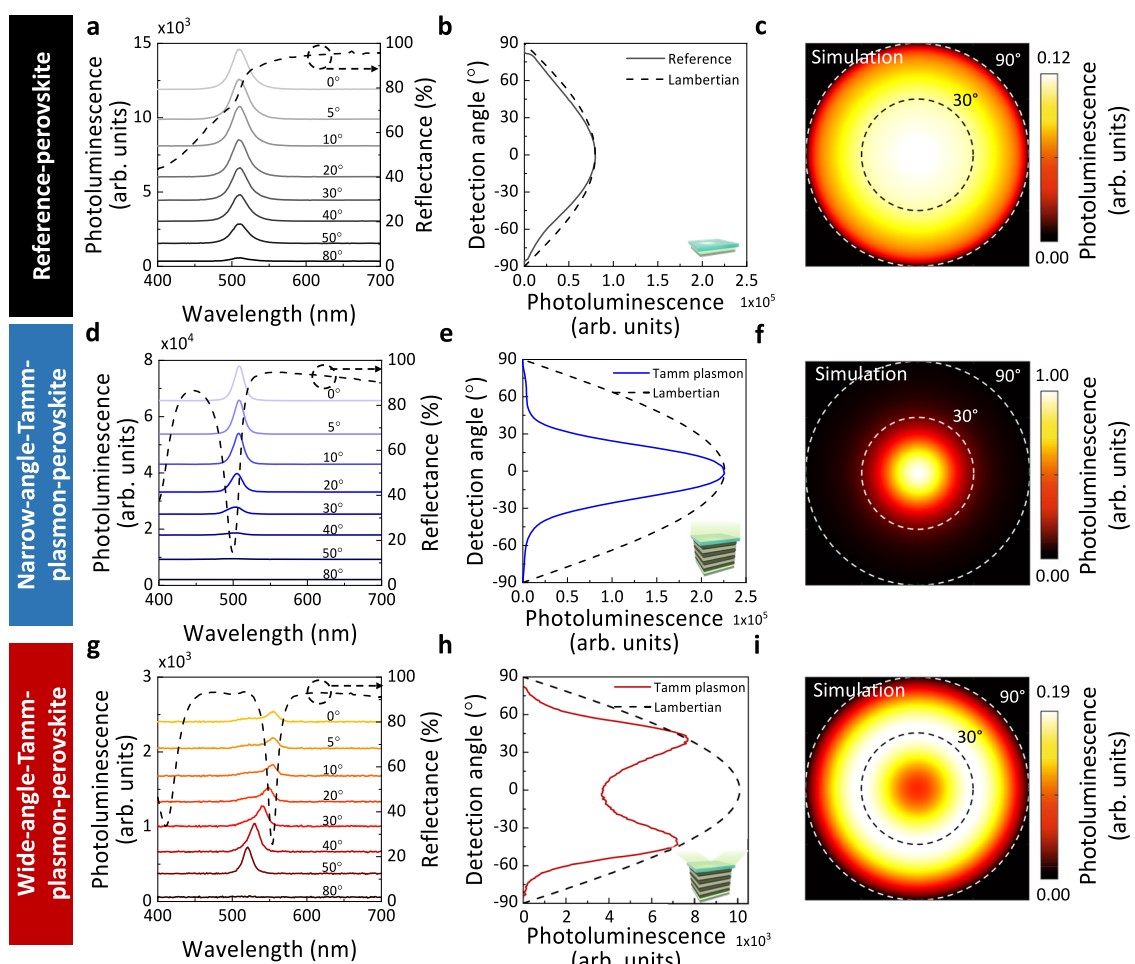

**Fig. 2 | Perovskite-based Tamm plasmon structures show tunability of the emission angular distribution.** Experimental angular photoluminescence (PL) results and simulations of (**a**–**c**) reference perovskite stack, (**d**–**f**) narrow-angle-Tamm-plasmon-perovskite structure with quasi-2D perovskite thickness of 26 nm and (**g**–**i**) wide-angle-Tamm-plasmon-perovskite structure with quasi-2D perovskite thickness of 50 nm. **a**, **d**, **g** PL intensity of each samples collected at increasing angle of 0°, 5°, 10°, 20°, 30°, 40°, 50° and 80°. Reflectance is shown as a dashed line. **b**, **e** Detection angle against PL of reference-perovskite stack and narrow-band-Tamm-plasmon-based perovskite structure with PL spectra integrated from 480 nm to 540 nm. **h** Detection angle against PL of wide-angle-Tamm-plasmon-perovskite structure with PL spectra integrated between 505 nm to 565 nm. Lambertian curve in dashed line. Inset: schematic of each structure. Since all samples are excited at the same power density, the integrated PL are comparable among the three samples. **c**, **f**, **i** Finite-difference time-domain simulation of angular output power of each structure. PL intensity of all configurations normalised to the PL intensity of narrow-angle-Tamm-plasmon-perovskite structure (see colour bars).

measured at small angles (dashed line). As the angle of emission increases, the PL blue-shifts in response to the dispersion relation of the Tamm plasmon mode[27]. The highest PL intensity is achieved when the blue shifting of the Tamm plasmon resonance matches the perovskite PL position, which is observed at a collection angle of 40° for this sample (Fig. 2h). We hereafter refer to these samples with wide-angled directionality as the wide-angle-Tamm-plasmon-perovskite structure. The increase of emission directionality angle is dependent on the increase of perovskite thickness, which leads to an increase in Tamm plasmon resonance wavelength at 0°. The large angle emission directionality of the wide-angle-Tamm-plasmon-perovskite structure shown experimentally in Fig. 2h matches the simulation shown in Fig. 2i. Results for the angle-dependent PL spectra for all wide-angle-Tamm-plasmon-perovskite structures can be found in Supplementary Fig. 9, demonstrating the versatility of this photonic platform.

## Tamm-plasmon-driven perovskite LEDs

To demonstrate the effect in electroluminescence from full devices, we fabricate PeLEDs with the structure of substrate/indium tin oxide (ITO)/ poly(4-butyltriphenylamine) (poly-TPD)/PVK/quasi-2D perovskite/2,2′,2″-(1,3,5-benzinetriyl)-tris(1-phenyl-1-H-benzimidazole)

(TPBi)/8-quinolinolato lithium (LiQ)/Ag. Here, the substrate for reference PeLEDs and Tamm-plasmon-driven PeLEDs refers to glass and glass/photonic crystal substrate respectively. The hole injection layers were formed by spin-coating layers of poly-TPD and PVK with thickness of ~10 nm and ~5 nm respectively. A 40-nm layer of thermally evaporated TPBi acts as the electron injecting layer, followed by evaporated LiQ (3 nm)/Ag (100 nm) as electrodes. Each of the device stack layers are optimised to be active parts of the photonic crystal/metal combinations to realise the Tamm plasmon effects (see Supplementary Fig. 10).

We observe strong EL spectral narrowing from FWHM of 22.4 nm in the reference PeLEDs to FWHM of 12.1 nm in the narrow-angle-Tamm-plasmon-driven PeLEDs (perovskite thickness ~26 nm) due to the sharp Tamm plasmon resonance matching the EL peak, as shown by the dashed line in Fig. 3a. The current density–voltage curves of the reference PeLEDs and narrow-angle-Tamm-plasmon-driven PeLEDs are similar (Fig. 3b), which is consistent with the photonic crystal layers being smooth and uniform, and lying below the ITO contact, thus not electrically modifying the PeLED device. The highest forward luminance of the narrow-angle-Tamm-plasmon-driven PeLEDs reached 21,800 cd m$^{-2}$ which is significantly higher than the reference PeLEDs

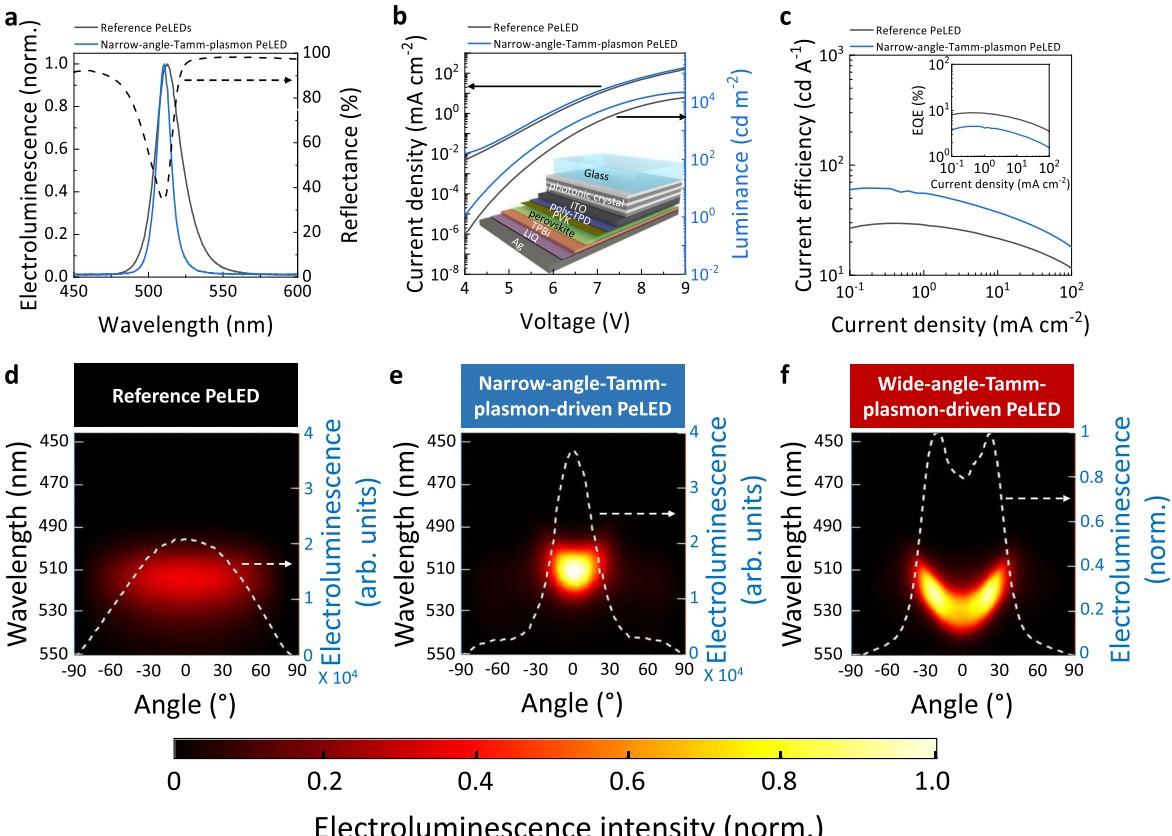

**Fig. 3 | Tamm-plasmon-driven PeLEDs with controlled directionality and electroluminescence intensity. a** Normalised electroluminescence (EL) spectrum of reference PeLEDs and narrow-angle-Tamm-plasmon-driven PeLED. Tamm plasmon resonance of the narrow-angle-Tamm-plasmon-driven PeLED shown with a reflectance measurement (black dashed line). **b** Current density–voltage–luminance performance of reference PeLED and narrow-angle-Tamm-plasmon-driven PeLED. Inset: schematic of Tamm-plasmon-driven PeLED. **c** Current efficiency–current density curve. Inset: EQE–current density curve considering emission over all angles. Angle-dependent EL spectrum of (**d**) reference PeLED, (**e**) narrow-angle-Tamm-plasmon-driven PeLED and (**f**) wide-angle-Tamm-plasmon-driven PeLED collected across the PeLED surface macroscopically. White dashed line: EL as a function of angle integrated from 490 nm to 540 nm (**d**, **e**). Normalised EL integrated from 490 nm to 540 nm in (**f**). EL intensity of all configurations each normalised to the EL intensity of narrow-angle-Tamm-plasmon-driven PeLED (see colour bar).

with a maximum of 14,500 cd m$^{-2}$. For applications such as displays where forward directional emission is crucial, current efficiency defined as the ratio between forward luminance and current is a more relevant parameter to consider[42]. Figure 3c shows that the maximum current efficiency of our Tamm-plasmon-driven PeLEDs at 61.8 cd A$^{-1}$ is much higher than the reference PeLEDs at 29.6 cd A$^{-1}$. We note that whilst there is an enhancement in the EL in the forward direction for the Tamm plasmon PeLED, there is a reduction in total EQE as that quantity considers emission in all directions (inset Fig. 3c), which matches the drop in photoluminescence quantum efficiency (PLQE) in the Tamm plasmon PeLED compared with the reference PeLED (see Supplementary Note 2). This observation may relate to optical losses which are common in coupling systems but could be improved with more careful design and improved perovskite quality. Unlike the bulk perovskite counterparts, the quasi-2D PeLED shows stronger roll-on and roll-off in EQE against current density (more discussion in Supplementary Note 3). The PL lifetime of reference PeLED and narrow-angle-Tamm-plasmon-driven PeLEDs structures shows no significant changes (more discussion in Supplementary Note 4 and Supplementary Fig. 11).

The EL spectra as a function of angle are plotted in image maps in Fig. 3d–f. The EL of the reference PeLED consistently peaks at 513 nm across all angles and shows a Lambertian emission profile with angular FWHM of 143.9° (white dashed line, Fig. 3d). In Fig. 3e, the narrow-angle-Tamm-plasmon-driven PeLED shows enhancement by a factor of 1.8 in forward directional EL with narrowed angular FWHM of 36.6°

(white dashed line) compared to the reference PeLED. There is a small blue shift in the EL spectrum from 0° to 30° due to blue shifting of the Tamm plasmon modes at increasing angle. Further increase in collection angle leads to strong suppression of EL intensity as the Tamm plasmon modes decouple from the original EL of the quasi-2D perovskite. In Fig. 3f (white dashed line), we show wide-angled EL in the wide-angle-Tamm-plasmon-driven PeLED by employing thicker (~54 nm) perovskite films. The performance of the wide-angle-Tamm-plasmon-driven PeLED is shown in Supplementary Fig. 12. The red-shifted EL spectrum at small angles gradually blue-shifts with increasing collection angle due to shifting in Tamm plasmon resonance modes and the EL intensity peaks when Tamm plasmon modes resonate at the perovskite EL. This result again demonstrates the versatility of the emission platform.

## Discussion

The EL angular response of both the narrow-angle and wide-angle-Tamm-plasmon-driven PeLEDs resemble the PL angular response of the narrow-angle and wide-angle-Tamm-plasmon-driven structures, respectively. Thus, we have demonstrated the translation of the PL directional enhancement in the perovskite-based Tamm plasmon structure into directional emission from an operating Tamm-plasmon-driven PeLED. Besides, simulations also show the adaptability of the Tamm-plasmon-driven PeLEDs in the red and blue regimes (Supplementary Fig. 13) by fine tuning the thicknesses of each layer. This work demonstrates the emitting material as the bottom layer of the

photonic crystal at the metal-photonic-crystal interface in a planar Tamm plasmon system enhances directional photo-excited emission and can be integrated into an LED to show directional electroluminescence amplification. Importantly, the PeLED device stack and other photonic layers are all optically and/or electrically active layers in the Tamm plasmon structure in a monolithic fashion without use of further complicated optics.

In Supplementary Fig. 14, we compare our design with simulated and optimised metal-metal and distributed Bragg reflector (DBR)-DBR microcavities, showing a significantly stronger electric field enhancement (normalised to the Tamm plasmon enhancement) in the Tamm plasmon perovskite structure (at 515 nm: 1.00; at 533 nm: 1.00) compared with DBR-DBR microcavities (at 515 nm: 0.60; at 533 nm: 0.56) and metal-metal microcavities (at 515 nm: 0.93; at 533 nm: 0.80). To compare the versatility of Tamm plasmon and metal-metal cavities in the highest efficiency blue[43], green[8] and red[44] perovskite LED devices, both Tamm plasmon and metal-metal cavities are simulated based on the reported device structures (Supplementary Fig. 15). For all three examples, the Tamm plasmon structure shows better spectral narrowing (sharper resonance) and higher normalised integrated electric field intensity within the perovskite layer for all blue (1.0 in Tamm plasmon, 0.41 in metal-metal), green (1.0 in Tamm plasmon, 0.61 in metal-metal) and red (1.0 in Tamm plasmon, 0.92 in metal-metal) PeLEDs. Furthermore, the Tamm plasmon structure employs the exact device structure as reported, with only the electrically inactive photonic crystal lying below ITO; practical replacement of ITO with metal in metal-metal cavities is challenging due to wettability and sensitivity issues when solution-processing the perovskite device layers on a metal, and very few experimental demonstrations of working PeLEDs are reported in this architecture. Thus, the Tamm plasmon structure has more degrees of freedom for design considerations - the photonic crystal can be more carefully designed to achieve a stronger electric field enhancement for a given LED device by tuning the electrically inactive photonic crystal, for example by adding extra metal oxide layers again. By contrast, the electric field intensity values for the metal-metal cavities are near their upper limits due to limitations on device performance, metal choice and absorption loss of the metal. Compared with the DBR-DBR cavities, the metal in the Tamm plasmon structure is easier to fabricate compared with depositing DBR on top of perovskite films and can readily act as the electrode for the LED device, thus is easier to implement.

As Ag has the lowest ohmic loss in the visible regime compared to other common metals, further improvements might be achieved by alloying the Ag metal, adding passivation layers or spacing interlayers, to further supress quenching and inhibit ionic reactions of the perovskite with the Ag electrode[35,45]. We expect further improvements by increasing the photonic crystal layer pairs to increase the coupling strength (Supplementary Fig. 3d), by adding anti-reflective coating on the air/glass interface to improve outcoupling efficiency at wide angles, as well as through investigation on the interaction between waveguiding modes and emission modes that can further enhance the emission of the Tamm-plasmon-driven PeLEDs. In this proof-of-concept demonstration, we balanced the number of layers (complexity in manufacturing) with the final performance of our emitter and device.

We demonstrated Tamm-plasmon-driven PeLEDs with strong directional and enhanced emission. The bottom photonic crystal layer at the metal-photonic-crystal interface in a Tamm plasmon system is replaced with an emitting quasi-2D halide perovskite layer to strongly confine the Tamm plasmon modes across the entire perovskite layer. As a result, the narrow-angle-Tamm-plasmon-driven PeLED shows good directionality (angular FWHM of 36.6° compared to reference 143.9°), EL enhancement by 1.8 times at small angles, and EL spectral narrowing from 22.4 nm to 12.1 nm. The excellent adaptability of Tamm-plasmon-driven PeLEDs covering the visible spectrum, strong

enhancement in forward direction, enhanced directional emission (with narrow-angle Tamm plasmon resonance), tuneability of large angle directional emission (with wide-angle Tamm plasmon resonance), EL spectral narrowing, small thickness of active structure and relatively simple scalability of fabrication methods makes Tamm-plasmon-driven PeLEDs desirable. The Tamm-plasmon-driven PeLEDs show potential in applications where angular control, emission intensity and colour purity are crucial, including displays, optical communications, virtual reality headsets, directional light sources, optical aligners and measuring systems.

## Methods
### Optical simulations
Optical simulations were conducted using a custom code based on the transfer matrix method[46] and a genetic algorithm for the Tamm plasmon structural design. After resolving the sample structure, the emission properties were analysed using a commercial finite-difference time-domain software (3D Electromagnetic Simulator from Lumerical Inc.)[47]. The refractive indices of all materials used in the simulations are detailed in Supplementary Fig. 16. This methodology involved isotropically embedding emitting dipoles (averaged over three perpendicular orientations) throughout the perovskite layer to examine the luminous power emitted from the structure's surface. Single-wavelength simulations were performed at wavelength, $\lambda = 514$ nm within a simulation box sized $2 \times 2 \times 1$ μm$^3$, with perfectly matched layers applied to all boundaries. The mesh grid dimensions were set to 10 nm for the x- and y-axis and 1 nm for the z-axis. A 2-dimensional frequency-domain field monitor was employed to capture the far-field projection via Fourier transform, capturing outgoing radiation across a hemisphere[48].

### Fabrication of 1-dimensional photonic crystal
One-dimentional photonic crystal made of 3 pairs of alternating titanium dioxide (TiO$_2$) and silicon dioxide (SiO$_2$) were deposited on glass substrates in the Institute of Optoelectronics, Military University of Technology. Thickness of each layer was TiO$_2$: ($70 \pm 2$) nm and SiO$_2$: ($90 \pm 2$) nm. The photonic crystal were deposited in an e-beam evaporation system with a plasma source assistance (Syrus 710 Pro, Bühler Leybold Optics, Alzenau, Germany). The base pressure of the system was $2 \times 10^{-6}$ mbar. TiO$_2$ and SiO$_2$ were deposited at 0.25 nm s$^{-1}$ and 0.6 nm s$^{-1}$ rate, respectively.

### ITO sputtering
ITO electrodes were sputtered on glass substrates (for reference PeLEDs) and photonic crystal substrates (for Tamm-plasmon-driven PeLEDs) using a custom setup located in the Class 10,000 clean room in the Electrical Engineering Division, Department of Engineering, University of Cambridge. The ITO electrodes were patterned using a metal mask. The ITO sputtering utilised an In$_2$O$_3$/SnO$_2$ 90/10 wt% target, operated at argon flow of 20 sccm, with the pressure of 5 mTorr and power of 40 W. The sputtering rate achieved was 3.7 nm minute$^{-1}$, resulting in ITO conductivity of 1800 S cm$^{-1}$ as measured by a 4-point-probe, with an ITO thickness of 83 nm determined by AFM.

### Materials
Lead (II) bromide (PbBr$_2$, 99.999%), phenylethylammonium bromide (PEABr, >99.5%), 1,4,7,10,13,16-hexaoxacyclooctadecane (18-crown-6, ≥99%), poly(4-butyltriphenylamine) (poly-TPD, Mw ≥20,000 g mol$^{-1}$), poly(9-vinylcarbazole) (PVK, MW:25,000-50,000 mg mol$^{-1}$), dimethyl sulfoxide (DMSO, anhydrous, 99.9%), chlorobenzene (CB, anhydrous, 99.8%) were purchased from Sigma-Aldrich. Cesium bromide (CsBr, 99.999%) was purchased from Alfra Aesar. 2,2',2"-(1,3,5-Benzinetriyl)-tris(1-phenyl-1-H-benzimidazole) (TPBi, >99.5%), 8-Hydroxyquinolinolato-lithium (LiQ, >99%) were purchased from Ossila. All chemicals were used without any further purification.

## Preparation of perovskite precursor solution

Perovskite precursor was prepared by dissolving $PbBr_2$, CsBr and PEABr (molar ratio of 1: 1.05: 0.4) in DMSO at 0.25 M initially and then further diluted down to concentrations between 0.13 and 0.23 M for thickness variation. 18-crown-6 was added as additive (molar ratio of 1.7% to $PbBr_2$) in perovskite precursors for PeLEDs to improve the PeLED performance[49].

## Fabrication perovskite structures

Glass substrates and glass/photonic crystal substrates were cleaned using detergent, deionised water, acetone and isopropanol under ultrasonication for 10 minutes each, followed by a 15-min UV ozone treatment. A solution of PVK (6 mg ml$^{-1}$ in CB) was spin-coated onto the substrate at 4000 rpm for 30 s, then immediately annealed at 100 °C for 10 min. It is worth noting that other common hole injection layer such as Poly(3,4-ethylenedioxythiophene)-poly(-styrenesulfonate) (PEDOT:PSS), Poly(4-butyltriphenylamine) (poly-TPD), Poly[(9,9-dioctylfluorenyl-2,7-diyl)-co-(4,4′-(N-(4-sec-butylphenyl)diphenylamine)] (TFB) posses similar refractive indices, making them compatible with such optical structures. Perovskite precursors, with concentration ranging from 0.13 M to 0.25 M, were spin-coated at 6000 rpm for 90 s and immediately annealed at 70 °C for 5 min. A 100 nm thick Ag film was subsequently thermally evaporated onto the perovskite film at a rate of 0.1 nm s$^{-1}$.

## Fabrication perovskite LEDs

Glass/ITO substrates and glass/photonic crystal/ITO substrates were cleaned in detergent, deionised water, acetone and isopropanol under ultrasonication for 10 minutes each, then treated with UV Ozone for 15 min. Poly-TPD (10 mg ml$^{-1}$ in CB) and PVK (6 mg ml$^{-1}$ in CB) is sequentially spin-coated onto the substrate at 4000 rpm for 30 s and immediately annealed at 100 °C for 10 min and 140 °C for 20 min, respectively. The perovskite precursor of concentration 0.16 M (narrow-angle-Tamm-plasmon PeLED) and 0.25 M (wide-angle-Tamm-plasmon PeLED) was spin coated at 1000 rpm for 5 s and 4000 rpm for 55 seconds and then immediately annealed at 90 °C for 10 minutes. TPBi (40 nm), LiQ (3 nm) and Ag (100 nm) were then sequentially thermal evaporated on the perovskite film.

## Film thickness measurement

Film thicknesses were assessed by scanning across the depth of a scratch made on the film with a razor blade (Supplementary Fig. 4). The scan was performed on Bruker Dimension Icon atomic force microscope (AFM) with a silicon tip on Nitride lever (Bruker Scanasyst-Air cantilever, spring constant 0.4 N m$^{-1}$) running on peak force tapping mode. Data were analysed with WSxM 5.0 software[50].

## Reflectance measurement

Tamm plasmon resonance (reflectance) was characterised by UV−visible spectrometer (Shimadzu UV-3600Plus) with an integrating sphere attachment (Shimadzu ISR-603). The total reflectance was measured at 8° offset to keep the specular reflected light within the integrating sphere. Given that the reflectance of the Tamm-plasmon-perovskite structure shown in Fig. 1d was measured at 8°, the actual Tamm plasmon resonance wavelength at 0° should be ca. 3 nm red-shifted. The baseline measurement was done with a 100 nm thick evaporated silver mirror as a reference due to high reflectivity of our samples.

Reflectance measurement was cross-checked with the Agilent Cary7000 Universal Measurement Spectrometer using the Universal Measurement Accessory. The sample is tilted at 6° and the detector at 12° to collect the specular reflectance without blocking the excitation lamp. The baseline measurement is done at 100% transmittance without any reference, thus eliminating any inaccuracy due to the reference defects.

Microscale reflectance of photonic crystal substrates was measured with hyperspectral microscope (Photon Etc. IMA). A lamp light was focussed on the sample through a condenser from below the sample and was collected by the objective lens (Olympus MPLFLN20x with NA of 0.45) and measured by a CCD camera. The measured reflectance of photonic crystal was calibrated with reflectance of a calibration mirror.

## Photoluminescence quantum efficiency

PLQE of perovskite films, reference structures and Tamm plasmon structures are measured with a 405 nm continuous wave laser under excitation between 3 to 167 mW cm$^{-2}$ in an integrating sphere and the spectrum collected with Andor iDus Si detector. Calculations are based on 3 configurations of the sphere—empty sphere, sample placed in the sphere but laser beam directed on the sphere wall and laser beam directed onto the sample[51].

## Characterisation of perovskite LED performance

All PeLEDs were encapsulated in a $N_2$-filled glovebox with UV-cured resin and glass. The PeLEDs were measured under ambient condition with the LED measurement setup from the Optoelectronics group in Cavendish Laboratory. The PeLEDs were powered by a Keithley 2400 source metre as a voltage source for measuring the current density−voltage characteristics. The photon flux was simultaneously measured using a calibrated circular silicon photodiode centred over the light-emitting pixel. The luminance of the PeLEDs were calculated based on the emission function of the PeLEDs and on the known spectral response of the silicon photodiode. The EL spectra of the devices were measured using a Labsphere CDS 610 spectrometer. The current efficiency was calculated as the ratio between forward luminance and current. The resulting EQE was calculated considering the angular resolved emission using equation from Archer et al.[42].

$$\eta_{EQE}(V) = 100 \frac{2\pi r^2}{A_{PD}} \frac{V_{PD}(V)}{R_{PD}} \frac{q}{hc} \frac{1}{I_{PeLED}(V)} \frac{\int S(\lambda,0)\lambda d\lambda}{\int S(\lambda,0)R(\lambda)d\lambda} \int_0^{\frac{\pi}{2}} \frac{\int S(\lambda,\theta)\lambda d\lambda}{\int S(\lambda,0)\lambda d\lambda} \sin\theta \, d\theta \quad (1)$$

where $V_{PD}$, $R_{PD}$, $A_{PD}$ are voltage, resistance and area of photodiode respectively. $r$ is the distance between photodiode and PeLED. $S(\lambda,\theta)$ is the measured spectral radiant intensity at angle $\theta$. $R(\lambda)$ is the photodiode responsivity. $I_{PeLED}$ is the current across the PeLED. $q$, $h$, and $c$ are unit charge, Planck constant and speed of light respectively.

## Angular luminescence measurement

The macroscale angular PL and EL was measured on a home-built setup (Supplementary Fig. 17). For PL, a 405 nm continuous wave laser and the sample were fixed on the optical table with the excitation angle normal to sample surface in x−y direction and 20° above in the z-direction. The laser beam was focussed on the sample with a lens ($f$ = 1000 mm, power density of 0.5 W cm$^{-2}$). For EL, the PeLEDs were powered by the Keithley 2400 source meter at a constant current density of 0.44 mA cm$^{-2}$. Both PL and EL spectra was captured by a spectroscopy camera (Andor iDus DU420A Si detector) connected to a fibre collimator (Thorlabs F220SMA-532) through an optical fibre. The fibre collimator was fixed on an automated rotating stage (Thorlabs PRMTZ8 Motorised continuous rotation stage, Thorlabs K-Cube DC servo motor controller). The fibre collimated was fixed at 10 cm away from the sample to maximise the angular resolution while ensuring a high collection of fluorescence. The PL and EL spectra were collected at 1° interval with a scan rate of 3° s$^{-1}$ and 5° s$^{-1}$, respectively. In Supplementary Fig. 18, we showed the PL and EL stability are suitably stable over the total angular PL and EL measurement time scales for 60 s and 36 s, respectively, under the same continuous excitation and driving current of 0.5 W cm$^{-2}$ for angular PL and 0.44 mA cm$^{-2}$ for angular EL. In addition to the PL and EL stability, the PL and EL are always collected from one end to the

other end, e.g. from −90° to 90°. Thus, we make sure that the angular measurements are not affected by degradation by checking that the curves are symmetrical (see Figs. 2b, d, h and 3d–f). For both PL and EL measurements, all 4 edges of the samples were masked with black tape to eliminate emission from the sample edge.

The emission across the solid angle is calculated from the angular PL and EL measurements based on the equation shown below, which is derived from Archer et al.[42].

$$I_{\text{solid angle}} = I_0 \times \int_{-\alpha}^{\alpha} \frac{\int S(\lambda, \theta)\lambda d\lambda}{\int S(\lambda, 0)\lambda d\lambda} \sin\theta \, d\theta \qquad (2)$$

where $\alpha$ is the solid angle of interest, $I_{\text{solid angle}}$ is the PL or EL intensity, $I_0$ is the forward intensity, $S(\lambda, \theta)$ is the spectral radiant intensity.

Microscale PL was measured with hyperspectral microscope (Photon Etc. IMA), excited with a 405 nm continuous laser. The hyperspectral microscopy measurements were collected with objective lenses (Olympus MPLFLN) of 20x and NA of 0.45 (equivalent to collection angle of 26.7°).

### Transient photoluminescence

Time-resolved photoluminescence was measured at fluences between 1 and 370 nJ cm$^{-2}$ pulse$^{-1}$ with confocal microscope (PicoQuant MicroTime 200). The samples were excited with 405 nm pulsed laser (pulse width ~100 ps, repetition rate 5 MHz) that was focussed with an 10x air objective lens.

Time-resolved photoluminescence at lower fluence between 0.01 and 5 nJ cm$^{-2}$ pulse$^{-1}$ were measured with photoluminescence spectrometer (Edinburgh Instruments FLS1000). A 405 nm pulsed laser (pulse width ~50 ps, repetition rate 2 MHz) was focussed on the samples at 45° and 60° and the emission was collected between 45° and 30° respectively at 1 nm bandwidth. Time resolved emission scans were done by sweeping the emission collection wavelengths between 490-520 nm with 5 nm step. Measurements are done with both unencapsulated and encapsulated samples in air and both shows similar results.

### Cross-section of Tamm-Plasmon-perovskite structure

The TEM lamella cross-section was prepared with an FEI Helios Nanolab Dualbeam FIB/SEM following a standard protocol[52]. The lamella was transferred minimising air exposure into an FEI Osiris TEM operating at 200 kV and ~140 pA beam current. HAADF images were acquired using a Fischione detector at a camera length of 115 mm, with a dwell time of 1.9 μs and a spatial sampling of 0.7 nm pixel$^{-1}$. STEM-EDX maps were acquired using a Bruker Super-X silicon drift detector with a collection solid angle of ≈0.9 sr, a dwell time of 50 ms, a spatial sampling of 5 nm pixel$^{-1}$, and a spectral resolution of 5 eV channel$^{-1}$. STEM-EDX compositional maps were spectrally rebinned to 10 eV per channel, and denoised using PCA/NMF taking the first 8 components. Data processing was done using HyperSpy v1.6.1, a Python-based analysis suite for hyperspectral data[53].

## Data availability
The data that support the findings of this study are openly available in Apollo−University of Cambridge Repository at https://doi.org/10.17863/CAM.109169.

## Code availability
Details on the code that supports the findings of this study can be shared under request.

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

## Acknowledgements

The authors acknowledge the European Research Council (ERC) under the European Union's Horizon 2020 research and innovation program (HYPERION, grant agreement No. 756962), Cambridge Royce facilities grant EP/PO24947/1, Sir Henry Royce Institute—recurrent grant EP/R00661X/1 and the Engineering and Physical Sciences Research Council (EPSRC) (grant agreement Nos. EP/R023980/1, EP/T02030X/1 and EP/S030638/1). This work was co-financed by Military University of Technology under research project UGB 502-6700-23-759. Z.Y.O. acknowledges scholarship from St John's College, University of Cambridge. A.J.-S. gratefully acknowledges a postdoctoral scholarship from the Max Planck Society and the Spanish Ministry of Universities for funding through a Beatriz Galindo Research fellowship BG20/00015. K.G. appreciates support from the Polish Ministry of Science and Higher Education within the Mobilnosc Plus program (grant no.1603/MOB/V/2017/0) and the National Science Centre (2022/47/D/ST5/03332). Y.S. acknowledges CSC Cambridge Scholarship. J.F.O and C.D. acknowledge funding from the Engineering and Physical Sciences Research Council (EPSRC) Nano Doctoral Training Centre (EP/L015978/1). K.F. acknowledges a George and Lilian Schiff Studentship, Winton Sustainability Fund Studentship, the Engineering and Physical Sciences Research Council (EPSRC) studentship. H.S. thanks the UK Engineering and Physical Sciences Research Council (EPSRC) grant EP/S023046/1 for the EPSRC Centre for Doctoral Training in Sensor Technologies for a Healthy and Sustainable Future. S.Kahmann is grateful for funding from the German Academic Exchange Service (DAAD) (91793256) for a short-term research fellowship, and from the Leverhulme Early Career Fellowship funded by the Leverhulme Trust (ECF-2022-593) and the Isaac Newton Trust (22.08(i)). G.V. acknowledges the support of the Spanish Ministry of Education, Vocational Training and Sports through a Beca de Colaboración (Grant No. 23CO1/000162). M.A. acknowledges funding from the Leverhulme Early Career Fellowship (grant agreement No. ECF-2019-224) funded by the Leverhulme Trust and the Isaac Newton Trust and from the Royal Academy of Engineering under the Research Fellowship programme. M.A. and G.V. acknowledge support from MICIU/AEI/10.13039/501100011033 and the European Union NextGenerationEU/PRTR through a PID2022-142525OA-I00 grant and a Ramón y Cajal Fellowship (RYC2021-034941-I). S.D.S. acknowledges the Royal Society and Tata Group (grant no. UF150033). We thank Youcheng Zhang (Cavendish laboratory, University of Cambridge) for ITO conductivity checks. For the purpose of open access, the authors have applied a Creative Commons Attribution (CC BY) licence to any Author Accepted Manuscript version arising from this submission.

## Author contributions

Z.Y.O., A.J.-S., K.G., M.A. and S.D.S. conceived and developed the Tamm-plasmon-driven PeLEDs. A.J.-S. and G.V. modelled and optimised the Tamm-plasmon-driven PeLEDs with input from M.A. Z.Y.O. and Y.S. fabricated and optimised the perovskite-based Tamm plasmon structure. Z.Y.O fabricated and optimised the Tamm-plasmon-driven PeLEDs including ITO sputtering. Z.Y.O collected and analysed the AFM data, UV-visible spectroscopy data, angular PL and EL data. J.F.O. collected and analysed the HAADF-STEM and EDX data. K.F. performed the hyperspectral microscopy. H.S., S.Kahmann and Z.Y.O collected and analysed the PL decay data. Z.Y.O measured and analysed the PeLEDs performance. P.N. and M.P.N. prepared the photonic crystal. S.N.

measured the PL stability. S.Kar worked on solution-processed photonic crystal to study Tamm plasmon emission enhancement in early stage. S.D.S., M.A., B.V.L., N.C.G., C.D., P.N. and S.M. supervised the work undertaken in their laboratories. Z.Y.O, M.A. and S.D.S wrote the manuscript with comments from all the authors.

## Competing interests

The authors declare no competing interests.
