## [Peer Review File · Nature Communications]

Strong angular and spectral narrowing of electroluminescence in an integrated Tamm-plasmon-driven halide perovskite LEDREVIEWER COMMENTS

Reviewer #1 (Remarks to the Author):

The manuscript by Ooi et al. describes the introduction of a thin perovskite into a Tamm-Plasmon-based microcavity structure. As the perovskite layer is part of the photonic crystal, tuning its thickness shifts the TP mode, in turn resulting in a spectral shift and in particular a shift in the angle-dependent emission characteristics as a consequence of coupling a narrowband emitter into a shifting cavity. In turn, the authors describe a narrow-angle-emitting and wide-angle emitting variation of their TP stack. Furthermore, a full PeLED stack is introduced into the TP cavity, showing that the tuning of angle-dependent characteristics can be transferred to an active device with reasonable efficiency. The results are presented mostly clearly and sensibly. Before being able to recommend this for publication, I would have a few questions however.

1) I find the results of the paper to be clear but not very surprising. Coupling a narrow emitter to a cavity mode will always result in the observed characteristics and has been demonstrated before. The main novelty of the paper seems to come from the fact that the emitter is a perovskite and the cavity is based on a Tamm Plasmon resonance. While this works well, I suspect similar results could have been obtained in simpler metal-metal cavities, although maybe with a slightly lower Q factor. It would be nice to see a comparison (even just in simulation) of this TP cavity with a metal-metal cavity and a DBR-DBR cavity, showing the potential unique advantages of using the TP, if there are any.

2) To me the use of the term "quasi-2D" perovskite is confusing. I admit to not being an expert in 2D or quasi 2D perovskite, but can a film of up to 50nm thickness still be considered quasi 2D? How many monolayer stacks n would that represent? The authors also do not demonstrate any confinement-induced properties of their perovskite film, as I would expect from a low- n quasi 2D system.

3) The authors describe an overall emission enhancement in PL for their narrow angle structure compared to their reference and ascribe this effect to an enhanced reabsorption and reemission of the perovskite at 510nm. Is the PLQY of the perovskite so high that this is a sensible effect? I would usually consider an enhanced reabsorption to be mostly detrimental to emission efficiency. In the same context, the authors describe illuminating the samples at 405nm with similar power density. However, the total absorption of the excitation light could differ depending on the optical structure. In Figure S5, the authors show a simulated absorption profile, although to me it is not clear if this is comparable. Looking at the electric field simulation in Fig 1 and S2, there seems to be some field enhancement at 405nm in the TP structure, which could lead to increased absorption. This should be looked at in more detail.

4) The transient PL of the PeLED stack shows a strong increase in lifetime (I assume due to the inclusion of a spacer between perovskite and metal reducing quenching?). This is not mentioned in the manuscript and should be discussed in a bit more detail. In particular, why does the lifetime increase going from the reference without Ag to the reference with Ag?

5) In the cavity devices, I would have expected a shortened lifetime due to the Purcell effect. Why is that not the case here?

6) I would prefer to see the current efficiency and EQE vs Luminance or Current density

rather than Voltage. Also, the jV characteristics and efficiency of the wide-angle-structure is missing. The devices shown seem to exhibit both a strong roll-on as well as roll-off in efficiency. Is this intrinsic to the perovskite or a consequence of the stack design?

Reviewer #2 (Remarks to the Author):

In this work, the authors fabricated 2D perovskites-based light-emitting diodes (LEDs) on a 1D photonic crystal to achieve Tamm-plasmon-driven directional light amplification. Although Tamm plasmons have been reported before, it was the first time to utilize perovskite LEDs as active component of the Tamm plasmon structure together with the photonic crystals. Consequently, it enabled significant electroluminescence (EL) enhancement in forward direction, and very narrow and tunable EL angular dispersion. It provided an efficient approach to control the intensity and angular dispersion of EL emission in PeLEDs, which is attractive for wider applications in displays and lighting. The authors have done a systematic study on both the photoluminescence (PL) and EL emission in the Tamm-plasmon structure, with certain theoretical explanation. However there are still a few issues that require more clarification. Hence, I suggested a Major revision of this manuscript before further consideration for acceptance. There are detailed questions and comments listed in the followings:

- (1) The HAADF-STEM image in Figure 1a is too dim to tell the Quasi-2D perovskites. Better image should be provided.
- (2) From both Figure 1d and 1e, it is very obvious that it is a perovskite thickness of 31 nm, rather than 26 nm claimed in the manuscript, that gives a matching Tamm plasmon resonance with the perovskite film PL wavelength (510 nm). Meanwhile, all the sample fabrication and discussion in the later manuscript are conducted for the 26 nm thick perovskites. What's the reason? If it is not a typo, all those measurements should be re-done for 31 nm perovskites.
- (3) In Figure 2d, the reflectance at around 550nm wavelength is over 100 %. What's the reason for it? Related discussion should be provided to avoid any misunderstanding.
- (4) As shown in Figure 3c, the narrow-angle-Tamm-plasmon-driven LED device showed significantly increased current efficiency, but 20% dropped EQE. Even though a possible light-outcoupling-related reason was provided by the authors in the manuscript, it was still not very convincing. Can the authors also provide PLQY results for both samples? Does it match well with the EQE drop phenomenon?
- (5) The angular PL and EL spectra measurement are conducted with a scan rate of $0.33^\circ/s$ and $5^\circ/s$, respectively, which means a long excitation (both photo- and electrical excitation) time. Is there a PL or EL intensity drop during the scanning? To avoid any influence on the angular results, PL and EL stability under continuous excitation should be provided.
- (6) The PL decay lifetimes for Tamm-plasmon-perovskite structure and reference-perovskite stack are 0.7 ns and 0.6 ns. I feel this difference is too small to confirm the increase in photo-recycling. Meanwhile, to eliminating the measurement uncertainty/error, IRF signal in the PL lifetime measurement should also be provided since the calculated lifetime is already very close to the equipment IRF.

Reviewer #3 (Remarks to the Author):

This work demonstrates perovskite LEDs in optical resonant structures with resulting spectral narrowing and angular dispersion. Novel claim is the coupling to Tamm-plasmons at the metal/Bragg mirror interface.

From the introduction to the manuscript: "Unlike conventional optical microcavities, the Tamm-plasmon-driven PeLED does not require a cavity or an optical spacer. In fact, we utilise the PeLED as active component of the Tamm plasmon structure together with the photonic crystals."

Maybe it is ignorance on the part of this reviewer, but every result shown in this manuscript appears indistinguishable from the physics of traditional microcavity LEDs. For example, this work by Dodabalapur et al. in 1996 shows an organic LED microcavity with a Bragg mirror and a metal mirror.

Journal of Applied Physics 80, 6954 (1996); <https://doi.org/10.1063/1.363768>

In that work, the broad line width of the organic emitter allows more of the dispersion to be populated by electroluminescence, in contrast to this manuscript where the narrow perovskite emission coupled to the narrow angular dispersion results in an angular narrowing.

If the authors wish to pursue publication, I would like to see convincing discussion of:

- 1) What is a Tamm Plasmon and how is it different from a microcavity mode
- 2) How does this work differ from other microcavity LED demonstrations, especially those employing one metal and one Bragg mirror, that show virtually the same spectral narrowing.

In other words, it is not clear how these are truly distinct results. If they are the same phenomenon described using different language, then it is hard to see significant novelty in the work.

REVIEWER REPORTS

Reviewer #1

The manuscript by Ooi et al. describes the introduction of a thin perovskite into a Tamm-Plasmon-based microcavity structure. As the perovskite layer is part of the photonic crystal, tuning its thickness shifts the TP mode, in turn resulting in a spectral shift and in particular a shift in the angle-dependent emission characteristics as a consequence of coupling a narrowband emitter into a shifting cavity. In turn, the authors describe a narrow-angle-emitting and wide-angle emitting variation of their TP stack. Furthermore, a full PeLED stack is introduced into the TP cavity, showing that the tuning of angle-dependent characteristics can be transferred to an active device with reasonable efficiency. The results are presented mostly clearly and sensibly. Before being able to recommend this for publication, I would have a few questions however.

We thank the reviewer for the positive comments on the work. We are pleased that the reviewer sees the merit in the angle-dependent and narrowband emission of the Tamm plasmon perovskite structure and PeLEDs in our manuscript, and we believe that the changes we highlight below address the reviewer's comments and have improved the quality of the manuscript.

Comment 1

I find the results of the paper to be clear but not very surprising. Coupling a narrow emitter to a cavity mode will always result in the observed characteristics and has been demonstrated before. The main novelty of the paper seems to come from the fact that the emitter is a perovskite and the cavity is based on a Tamm Plasmon resonance. While this works well, I suspect similar results could have been obtained in simpler metal-metal cavities, although maybe with a slightly lower Q factor. It would be nice to see a comparison (even just in simulation) of this TP cavity with a metal-metal cavity and a DBR-DBR cavity, showing the potential unique advantages of using the TP, if there are any.

We thank the reviewer for raising this point. As suggested by the reviewer, we have now performed and added simulations for the Tamm-plasmon-driven perovskite structure, the metal-metal cavity and the DBR-DBR cavity in Supplementary Fig. 4.

Supplementary Fig. 4 | Simulation of electric field intensity enhancement across the cross-section of a Tamm plasmon perovskite structure (a), a perovskite in DBR-DBR microcavity (b) and a perovskite in metal (10 nm) - metal (120 nm) microcavity (c). Blue lines show the reflectance of each system, which has been optimised for 30 nm of perovskite. d) Electric field enhancement at 510 nm wavelength for a metal (10 nm)-metal (120 nm) microcavity as a function of the perovskite layer thickness. Maximum confinement of metal-metal microcavity occurs at perovskite thickness of 65 nm and yields an electric field enhancement of 5.

We show that for the perovskite thickness studied (i.e. ~ 30 nm which is optimised to ensure efficient charge injection in the ultimate LED architecture), the system based on a Tamm plasmon structure presents the strongest electromagnetic field confinement. On the one hand, the metal-metal case does not have the ability to achieve resonance at a frequency close to the emission maximum of the active material when using a thickness of ~ 30 nm that is optimal for LED performance (see Supplementary Fig. 4c). To exemplify this, Supplementary Fig. 4d shows an electromagnetic field intensity map for a metal (10 nm) - metal (120 nm) cavity. We note that in the metal-metal cavity, the top metal layer is kept thin as it is limited by the trade-off between fundamental material absorption loss and reflectivity (DOI: 10.1093/oso/9780198782995.003.0001). Comparing the Tamm-plasmon-driven perovskite structure and the metal-metal cavity, we observed higher transparency loss in the metal-metal cavity even with a 10 nm thick silver layer. Furthermore, from Supplementary Fig. 4d, it is clearly seen how the maximum confinement occurs for perovskite thicknesses ~ 65 nm – which importantly would be detrimental to charge injection in LEDs. Unlike in the Tamm plasmon structure where we can design the photonic crystals and optimise the perovskite thickness to fit both good electric field confinement and LED performance, the metal-metal cavity has less freedom in design parameters. Nevertheless, we note a significant decrease in the electric field enhancement from a factor of 34 in our Tamm-plasmon-driven perovskite structure to around 2 (or a maximum of 5 at maximum confinement) in the metal-metal cavity when compared to non-structured references.

On the other hand, an optimised DBR-DBR cavity (see Supplementary Fig. 4b) shows a 7-fold enhancement in electric field when compared with the non-photonic/plasmonic reference, only slightly better than the metal-metal cavity though still significantly lower than the Tamm-plasmon-

driven perovskite structure. We note we restrict all the simulated DBRs to 3 pairs of bilayers in order to minimise complexity in the manufacturing and photonic-into-device integration processes though still preserving the desired colour purification and angular control. Critically, the Tamm plasmon structure is easier to implement especially as the metal layer of Tamm plasmon can act as an electrode for a device; special designs and patterned electrodes are not required.

We have added the following in the Discussion section of our main manuscript to stress the point:

“In Supplementary Fig. 4, we compare our design with simulated metal-metal and distributed Bragg reflector (DBR)-DBR microcavities, showing a significantly stronger electric field enhancement with the Tamm plasmon perovskite structure (Tamm plasmon: 34; metal-metal: 5; DBR-DBR: 7). In addition, compared with the metal-metal cavity, the Tamm plasmon structure has more degrees of freedom for design considerations - the photonic crystals can be carefully designed to achieve a strong electric field enhancement for a given LED device; the metal-metal cavities, however, have restricted freedom of perovskite thickness to trade-off between electric field enhancement and ideal LED design (see Supplementary Fig. 4d) and requires minimally thin top metal layer to reduce transparency losses. Compared with the DBR-DBR cavities, the metal in the Tamm plasmon structure is easier to fabricate compared with depositing DBR on top of perovskite films and can readily act as the electrode for the LED device, thus is easier to implement.”

Comment 2

To me the use of the term "quasi-2D" perovskite is confusing. I admit to not being an expert in 2D or quasi 2D perovskite, but can a film of up to 50nm thickness still be considered quasi 2D? How many monolayer stacks n would that represent? The authors also do not demonstrate any confinement-induced properties of their perovskite film, as I would expect from a low- n quasi 2D system.

We thank the reviewer for this important point and agree that the term “quasi-2D” perovskite caused some confusion. Quasi-2D perovskite is a general category of any perovskite with a mixed number of inorganic octahedral layers, n up to $n > 5$ layers sandwiched between the organic cation ligands (DOI: 10.1038/NNANO.2016.110). Typically, this refers to smaller ratios of small n layers and a majority of large n or bulk layers. Thus, film up to 50 nm is still possible.

Quasi-2D perovskite forms self-assembled multi-quantum-well structures to establish quantum and dielectric confinement (DOI: 10.1038/s41377-021-00501-0). The carriers funnel from smaller n (higher bandgap) to larger n (lower bandgap) and accumulate at the largest n recombination centres, thus increasing the confined carrier density and the radiative recombination. For the best quasi-2D perovskite LED performance, there is a trade-off between the carrier confinement and carrier transfer through the distribution of n which could be controlled by the amount of organic ligands added and/or the presence of additives to control the rate of reaction (DOI: 10.1007/s40820-022-00807-7). We found through optimisation that the PEABr ratio we use in this work shows the highest PLQY and EQE, which matches the literature (DOI: 10.1038/s41928-020-00487-4, DOI: 10.1038/s41467-018-06425-5 etc.).

We have now added a Supplementary Fig. 2 in which we show the absorbance and PL spectra of our quasi-2D perovskite. The absorbance of the quasi-2D perovskite film shows absorption peaks associated with the presence of low n phases in our material. However, the PL spectra show only a single narrow peak at the large- n bandgap which indicates efficient funneling of carriers from low- n to large- n as expected.

Supplementary Fig. 2 | Absorbance and PL spectra of quasi-2D perovskite film without the 18-crown-6 additive. Quasi-2D perovskite is a general category of any perovskite with a mixed number of inorganic octahedral layers, n up to $n > 5$ layers sandwiched between the organic cation ligands⁹. The carriers funnel from smaller n (higher bandgap) to larger n (lower bandgap), accumulate at the largest n recombination centres and show only single narrow photoluminescence peak at the large- n bandgap.

We have modified the Results section on Page 6 of the main manuscript as follows:

“This material offers high brightness, photoluminescence quantum yield and external quantum efficiency (EQE) when incorporated in LEDs due to dielectric and quantum confinement effects governed by the formation of lower dimensional halide perovskite structures^{36,37} (Supplementary Fig. 2).”

Where 36 and 37 are citations from Zhao *et al.* and Ban *et al.* respectively.

Comment 3

The authors describe an overall emission enhancement in PL for their narrow angle structure compared to their reference and ascribe this effect to an enhanced reabsorption and reemission of the perovskite at 510nm. Is the PLQY of the perovskite so high that this is a sensible effect? I would usually consider an enhanced reabsorption to be mostly detrimental to emission efficiency. In the same context, the authors describe illuminating the samples at 405nm with similar power density. However, the total absorption of the excitation light could differ depending on the optical structure. In Figure S5, the authors show a simulated absorption profile, although to me it is not clear if this is comparable. Looking at the electric field simulation in Fig 1 and S2, there seems to be some field enhancement at 405nm in the TP structure, which could lead to increased absorption. This should be looked at in more detail.

We thank the reviewer for this important comment. We have now measured the PLQY of our quasi-2D perovskite film (without 18-crown-6 additive) and got values around $5 \pm 1\%$. In addition, the PLQY of the reference perovskite (glass/PVK/quasi-2D perovskite/Ag) and Tamm plasmon perovskite structures are $< 1\%$ due to strong metal quenching. We agree that the PLQY is very low to claim the possibility of having photon-recycling.

We have now added the PLQE results in Supplementary Note 2 as follows:

“In Fig. 2, the reference and Tamm plasmon perovskite structure have quasi-2D perovskite without the 18-crown-6 as additive as the emitting material. The PLQE of the quasi-2D perovskite film without the 18-crown-6 additive is $5\pm 1\%$ at excitation fluence of 33 mW cm^{-2} . However, due to strong metal quenching, the PLQE of reference perovskite (glass/PVK/quasi-2D perovskite/Ag) and Tamm plasmon perovskite structure (glass/photonic crystals/PVK/quasi-2D perovskite/Ag) are both $<1\%$ at excitation fluence of 33 mW cm^{-2} .”

We have also added the PLQE procedures in Methods as follows:

“Photoluminescence Quantum Efficiency (PLQE). PLQE of perovskite films, reference structures and Tamm plasmon structures are measured with a 405 nm continuous wave laser under excitation between 3 to 167 mW cm^{-2} in an integrating sphere and the spectrum is collected with Andor iDus Si detector. Calculations are based on 3 configurations of the sphere – empty sphere, sample placed in the sphere but laser beam directed on the sphere wall and laser beam directed onto the sample ⁵¹.”

where 51 is the reference from De Mello *et al.*

As the perovskite is the only emitting material within the structure, only absorption of the laser at 405 nm within the perovskite layer will contribute to emission at 510 nm. In Supplementary Fig. 7 at an excitation wavelength of 405 nm, the total absorption within the perovskite layer at 0° (integrated absorption across each point of the perovskite) of the Tamm plasmon perovskite structure is 40%, which is relatively lower than the reference perovskite structure with absorption of 50%. Thus, the Tamm plasmon structure does not show increased absorption at the excitation wavelength of 405 nm.

Supplementary Fig. 7 | Simulated absorption profile of perovskite in Tamm-plasmon-perovskite structure. The absorption profile of perovskite layer at **a)** 405 nm (excitation wavelength), **b)** 510 nm (quasi-2D perovskite emission wavelength) and **c)** 514 nm in reference-perovskite and narrow-angle-Tamm-plasmon-perovskite structure.

We have now modified our claim on no increase in perovskite absorption at the excitation wavelength in the Results section on Page 8 of the main manuscript.

“The simulated absorption across the entire perovskite layer at the excitation wavelength of 405 nm within both the reference-perovskite structure and the narrow-angle-Tamm-plasmon-perovskite structure are comparable (Supplementary Fig. 7), thus showing the forward enhancement in PL is primarily due to the Tamm plasmon modes resonating at the emission wavelength of 510 nm rather than photon recycling by reabsorption-re-emission events.”

However, we note that we measure the angular PL across the xy-plane at a height perpendicular to the sample $z = 0$ as shown in Supplementary Fig. 17. In the original manuscript we integrate the area under the curve of Fig. 2b and Fig. 2e, which does not mean total emitted photons but rather just in the $z = 0$ plane. In fact, the solid angle needs to be considered. If we consider all emitted photons by considering the solid angle calculations (added details in Methods), we in fact do not see an overall enhancement of total emitted photons – the enhancement is specifically in the forward angular direction.

We have added the solid angle calculations in Methods.

“The emission across the solid angle is calculated from the angular PL and EL measurements based on the equation shown below, which is derived from Archer et al. ⁴⁴.

$$I_{solid\ angle} = I_0 \times \int_{-\alpha}^{\alpha} \frac{\int S(\lambda, \theta) \lambda d\lambda}{\int S(\lambda, 0) \lambda d\lambda} \sin \theta d\theta$$

where α is the solid angle of interest, $I_{solid\ angle}$ is the PL or EL intensity, I_0 is the forward intensity, $S(\lambda, \theta)$ is the spectral radiant intensity.”

As the highlight of the work is about controlling emission directionality, narrowing spectrum linewidth and enhancing forward emission instead of increasing the total emitted photons, we have now modified the discussion in the Results section on Page 8 of the main manuscript to remove the possible false claim on the photon-recycling as below.

“As shown in Fig. 2e, the narrow-angle-Tamm-plasmon-perovskite exhibits sharp emission directionality with angular FWHM of 44.7° compared to 124.5° for the reference perovskite with the same excitation power density (excited with a 405 nm laser). While all samples were excited at the same excitation intensity, the narrow-angle-Tamm-plasmon-perovskite shows strong enhancement by a factor of 2.6 in forward emission within a solid angle of $\pm 15^\circ$ perpendicular to the sample surface (calculations in Methods), which is crucial in applications that require forward emission.”

Comment 4

The transient PL of the PeLED stack shows a strong increase in lifetime (I assume due to the inclusion of a spacer between perovskite and metal reducing quenching?). This is not mentioned in the manuscript and should be discussed in a bit more detail. In particular, why does the lifetime increase going from the reference without Ag to the reference with Ag?

We thank the reviewer for raising this point. In Supplementary Fig. 12, the transient PL is measured on both the reference PeLED without Ag sample and reference PeLED with Ag sample, where both samples include the electron transport layer (TPBi/LiQ) on the perovskite films, thus change of PL lifetime due to quenching is less dominant (DOI: 10.1103/PhysRevB.55.7249).

As we found a small change in PL lifetime from sample to sample due to traps and defects, we repeated the transient PL measurements with repeated samples for each category and updated Supplementary Fig. 12 with more representative results. We observe similar transient PL for both the reference PeLED without Ag (yellow) and the reference PeLED (black) at the fast decay regime, however, the tail of the reference PeLED (black) shows a slightly longer lifetime. This could be due to the Ag layer that acts as

a good reflector which could vary the photon density within the perovskite film, thus affecting the lifetime (DOI: 10.1515/nanoph-2021-0067).

We have now added the discussion in Supplementary Note 3.

“In Supplementary Fig. 12, we compare the transient PL of a reference PeLED with and without Ag (black and yellow line, respectively) and a Tamm plasmon PeLED (green line). Firstly, as all structures include an electron transport layer (TPBi/LiQ), which acts as a spacer between the perovskite film and Ag, we observe very similar PL lifetime in all samples as the metal quenching becomes less dominant than in the cases shown in Supplementary Fig. 6³. Moreover, the PL lifetime tail becomes slightly longer for the PeLEDs with Ag, possibly due to a better reflection of light which could vary the photon density within the perovskite film⁴.”

Where 3 and 4 are cited references from Amos *et al.* and Raja *et al.* respectively.

Supplementary Fig. 12 | Transient PL. Time-resolved PL of reference PeLED without Ag, reference PeLED and Tamm-plasmon-driven PeLED with effective lifetimes of roughly 10 ns, 10 ns and 8.5 ns respectively. All perovskite films shown in this figure are quasi-2D perovskite with 18-crown-6 additive. Samples excited at 5 nJ/cm²/pulse with Edinburgh Instruments FLS1000. Effective lifetime is defined as the time required for PL intensity reduces to 1/e of the initial intensity.

Comment 5

In the cavity devices, I would have expected a shortened lifetime due to the Purcell effect. Why is that not the case here?

We thank the reviewer for this important comment and we agree that the Purcell effect needs consideration here.

In the original manuscript, we measure the PL lifetimes at different excitation and detection angles, fluences and emission wavelengths of a sample from each category. We now repeated the PL lifetime measurements with a few different samples in each category. We found that the lifetime varies slightly in each sample which could be due to variations in defects (traps) and sample-to-sample variation. In the updated Supplementary Fig. 6 and 12 (see previous comment), we report no significant change in PL lifetime in both the narrow-band-Tamm-plasmon perovskite structure (Supplementary Fig. 6) and narrow-band-Tamm-plasmon PeLED (device architecture, Supplementary Fig. 12) compared with their

respective reference perovskite structures – an effect we see as most representative of the samples. The no significant change in our Tamm plasmon PL lifetime is in agreement with several reports on planar confinement systems under a weak-coupling regime which shows less than 10% change (DOI: 10.1103/PhysRevLett.71.517, DOI: 10.1002/adom.201700523, DOI: 10.1039/D2QM00313A, DOI: 10.1063/1.116858).

We have now updated this point in the Results section on Page 8 of the main manuscript as follows.

“In Supplementary Fig. 6, we observe no significant changes in PL lifetime of the narrow-angle-Tamm-plasmon-perovskite compared with the reference perovskite structure considering small sample-to-sample variation, which is consistent with other planar confined systems operating under a weak confinement regime and reported very small <10% changes ^{34,35,42,43}. “

Where 34,35,42,43 are cited references from Vredenberg *et al.*, Lova *et al.*, Megahd *et al.* and Jordan *et al.*

Supplementary Fig. 6 | Transient PL. Time-resolved PL of reference-perovskite structure without Ag, reference-perovskite structure with Ag and Tamm-plasmon-perovskite structure with effective lifetimes of roughly 3.7 ns, 1.5 ns and 1.4 ns respectively. All perovskite films shown in this figure are quasi-2D perovskite without 18-crown-6 additive. Samples excited at 5 nJ/cm²/pulse with Edinburgh Instruments FLS1000. Effective lifetime is defined as the time required for PL intensity reduces to 1/e of the initial intensity.

Supplementary Fig. 12 | Transient PL. Time-resolved PL of reference PeLED without Ag, reference PeLED and Tamm-plasmon-driven PeLED with effective lifetimes of roughly 10 ns, 10 ns and 8.5 ns

respectively. All perovskite films shown in this figure are quasi-2D perovskite with 18-crown-6 additive. Samples excited at 5 nJ/cm²/pulse with Edinburgh Instruments FLS1000. Effective lifetime is defined as the time required for PL intensity reduces to 1/e of the initial intensity.

Comment 6

I would prefer to see the current efficiency and EQE vs Luminance or Current density rather than Voltage. Also, the jV characteristics and efficiency of the wide-angle-structure is missing. The devices shown seem to exhibit both a strong roll-on as well as roll-off in efficiency. Is this intrinsic to the perovskite or a consequence of the stack design?

We thank the reviewer for this comment. As requested by the reviewer, we have now changed Fig. 3c to current efficiency and EQE vs current density. In addition to that, we have updated the EQE after correcting more precisely for the angular dispersion of the emitted light (more details in Reviewer #2 Comment 4).

Fig. 3 | Tamm-plasmon-driven PeLEDs with controlled directionality and electroluminescence intensity. c) Current efficiency (CE)-J curve. Inset: EQE-J curve considering emission over all angles.

We have also added the JV characteristics and efficiency of the wide-angle-Tamm-plasmon PeLED in Supplementary Fig. 13.

Supplementary Fig. 13 | Wide-angle-Tamm-plasmon-driven PeLED performance. a) Current density-Voltage-Luminance (blue line). **b)** Current efficiency and EQE (inset) vs current density.

We have now added the discussion on roll-on and roll-off efficiency in quasi-2D perovskite LEDs in Supplementary Note 4:

“Similar to our observations in both reference PeLED and Tamm plasmon PeLED shown in Fig. 3c and Supplementary Fig. 13, quasi-2D PeLED has shown stronger roll-on and roll-off in efficiency compared to their 3D bulk counterparts. The stronger roll-on is mainly due to efficient funnelling and confinement of charge carriers in naturally formed quantum well-like structures with a mixed number of inorganic octahedral layers (see Supplementary Fig. 2); while the strong roll-off at high current density is likely due to charge injection imbalance, Auger-induced luminescence quenching and migration of organic ligands⁵⁻⁸. This is still a matter of investigation in the field, which is of our interest but not within the specific scope of this work.”

Where 5-8 are references from Ban *et al.*, Zou *et al.*, Warby *et al.* and Cheng *et al.* respectively.

Reviewer #2

In this work, the authors fabricated 2D perovskites-based light-emitting diodes (LEDs) on a 1D photonic crystal to achieve Tamm-plasmon-driven directional light amplification. Although Tamm plasmons have been reported before, it was the first time to utilize perovskite LEDs as active component of the Tamm plasmon structure together with the photonic crystals. Consequently, it enabled significant electroluminescence (EL) enhancement in forward direction, and very narrow and tunable EL angular dispersion. It provided an efficient approach to control the intensity and angular dispersion of EL emission in PeLEDs, which is attractive for wider applications in displays and lighting. The authors have done a systematic study on both the photoluminescence (PL) and EL emission in the Tamm-plasmon structure, with certain theoretical explanation. However there are still a few issues that require more clarification. Hence, I suggested a Major revision of this manuscript before further consideration for acceptance. There are detailed questions and comments listed in the followings:

We thank the reviewer for the positive comments on the work. We are pleased that the reviewer sees the merit in Tamm-plasmon-driven directional light amplification, and we believe that the changes we highlight below address the reviewer’s comments and have improved the quality of the manuscript.

Comment 1

The HAADF-STEM image in Figure 1a is too dim to tell the Quasi-2D perovskites. Better image should be provided.

We thank the reviewer for this comment and we agree that the quasi-2D perovskite looks dim in the image. However, since we do not want to modify the brightness of the original image, this is the best brightness we could achieve without overexposing the silver layer due to the limitations of the HAADF-STEM system.

In this technique, the STEM probe finely focuses high-energy electrons and scans across the thin sample where most electrons are diffracted through small angles, and some scattered through larger angles. The scattered beams at larger angles are collected by the annular detector for Z-contrast image (DOI: 10.1146/annurev.ms.22.080192.001131). The HAADF-STEM technique can only resolve

elements with Z-contrast above aluminium as a rule of thumb, however, the denser elements will dominate the signal by a power of 1.7 to the Z-contrast (DOI: 10.1146/annurev.ms.22.080192.001131). However, due to the presence of the silver layer in our Tamm plasmon perovskite structure, the brightest detection in HAADF-STEM is the heavy silver element. Thus, this limitation leads to other lighter layers including the perovskite layer appearing relatively dimmer. Our observation is inherent to the HAADF-STEM imaging.

Besides the Z-contrast limitation, we kept the electron dose low to minimise degradations of the soft perovskite and organic layers induced by strong electron beams. Thus, the HAADF-STEM image is slightly dim. Nevertheless, we can still clearly observe the contrast difference between each layer and even more apparent when comparing it with the EDX map.

Comment 2

From both Figure 1d and 1e, it is very obvious that it is a perovskite thickness of 31 nm, rather than 26 nm claimed in the manuscript, that gives a matching Tamm plasmon resonance with the perovskite film PL wavelength (510 nm). Meanwhile, all the sample fabrication and discussion in the later manuscript are conducted for the 26 nm thick perovskites. What's the reason? If it is not a typo, all those measurements should be re-done for 31 nm perovskites.

We thank the reviewer for this important comment. We note the Tamm plasmon resonance wavelength is measured with an integrating sphere where the sample is tilted at 8° normal to the excitation lamp. The 8° tilting is required to capture the specular reflectance within the integrating sphere. As we know that the Tamm plasmon resonance blue shifts with increasing angle, the actual Tamm plasmon resonance at 0° is estimated to be ca. 3 nm red-shifted (clarified in the Methods) compared to the results shown in Figure 1d and 1e. In Supplementary Fig. 10 (see below), we show that the Tamm-plasmon-perovskite structure with 31 nm thick perovskite film (resonance estimated to be 513 nm at 0°) shows a small angle broadening in PL compared with the Tamm-plasmon-perovskite structure with 26 nm thick perovskite film (estimated to be 504 nm at 0°), which demonstrates that a very small red-shifted Tamm plasmon resonance compared with the perovskite PL will increase the angle FWHM. Since we focus on showing sharp forward directional emission, we intentionally use 26 nm thick perovskite sample as the best parameter to display the effect of a Tamm plasmon resonance.

Supplementary Fig. 10 | Angular dependent PL spectra of Tamm-plasmon-perovskite structure. Normalized PL spectra collected across the Tamm-plasmon-perovskite sample surface with quasi-2D perovskite thickness of **a)** 26 nm (narrow-angle-Tamm-plasmon shown in Figure 2d-f), **b)** 31 nm, **c)** 36 nm, **d)** 41 nm, **e)** 45 nm and **f)** 50 nm (wide-angle-Tamm-plasmon shown in Figure 2g-i).

We have now clarified the 8° tilting in reflectance measurement in the figure caption and in the Method section as follows:

*“**Reflectance measurement.** The Tamm plasmon resonance (reflectance) was characterised by UV-visible spectrometer (Shimadzu UV-3600Plus) with an integrating sphere attachment (Shimadzu ISR-603). The total reflectance was measured at 8° offset to keep the specular reflected light within the integrating sphere. Given that the reflectance of the Tamm-plasmon-perovskite structure shown in Fig. 1d was measured at 8°, the actual Tamm plasmon resonance wavelength at 0° should be ca. 3 nm red-shifted. The baseline measurement was done with a 100 nm thick evaporated silver mirror as reference due to high reflectivity of our samples.”*

Additionally, we have added the following in the Results section on Page 6 of the main manuscript:

“We note that the Tamm plasmon resonance wavelengths from Fig. 1d and reported in Fig. 1e are measured using an integrating sphere where the sample is tilted at 8° (see Methods for details); the Tamm plasmon resonance at 0° is estimated to be ca. 3 nm red-shifted from this value. We thus found that a perovskite thickness of 26 nm gives the best matching to display the effects of a Tamm plasmon resonance with the perovskite film PL wavelength (dashed black line).”

Furthermore, we have added the 8° clarification in the axes of Figures 1d and 1e.

Fig. 1 | Development of the perovskite-based Tamm plasmon structure. **d)** Experimental optimization of perovskite-based Tamm-plasmon structure by varying the perovskite thicknesses between 20 nm to 54 nm (Tamm plasmon resonance wavelength best fitted with transfer matrix model to estimate thickness of perovskite). Tamm plasmon resonance dip acquired from reflectance measurements at 8°. PL of quasi-2D perovskite thin film on glass is shown as dashed line. **e)** Tamm plasmon resonance wavelength at 8° versus perovskite film thickness. Perovskite thickness is measured under AFM (solid line) and estimated from Tamm plasmon resonance wavelength best fitted with simulation (dashed line). Inset: schematic of the perovskite-based Tamm-plasmon structure with a green arrow indicating the quasi-2D perovskite layer.

Comment 3

In Figure 2d, the reflectance at around 550nm wavelength is over 100 %. What's the reason for it? Related discussion should be provided to avoid any misunderstanding.

We thank the reviewer for raising this point and agree that this could cause misunderstanding. The reflectance measurement is relative to a reference as the baseline. As silver reacts in the air, we suspect that our reflectance went over 100% due to an error in our imperfect silver reference.

We have repeated this measurement with a freshly evaporated silver reference and updated Figure 2d as shown below.

Fig. 2 | Perovskite-based Tamm plasmon structures show tunability of the emission angular distribution. Experimental angular PL results and simulations of **(d)** narrow-angle-Tamm-plasmon-perovskite structure with quasi-2D perovskite thickness of 26 nm. **(d)** PL intensity of each sample collected at increasing angle of 0°, 5°, 10°, 20°, 30°, 40°, 50° and 80° from top to bottom. Reflectance is shown as a dashed line.

We have also cross-checked the result with another UV-vis spectrometer which allows baseline measurement at 100% transmittance without any reference, thus eliminating the effect of defective reference. We observed similar reflectance as shown above with maximum reflectance of 95%.

We have now added the following text in the Methods section:

“Reflectance measurement was cross-checked with the Agilent Cary7000 Universal Measurement Spectrometer using the Universal Measurement Accessory. The sample is tilted at 6° and the detector at 12° to collect the specular reflectance without blocking the excitation lamp. The baseline measurement is done at 100% transmittance without any reference, thus eliminating any inaccuracy due to the reference defects.”

Comment 4

As shown in Figure 3c, the narrow-angle-Tamm-plasmon-driven LED device showed significantly increased current efficiency, but 20% dropped EQE. Even though a possible light-outcoupling-related reason was provided by the authors in the manuscript, it was still not very convincing. Can the authors also provide PLQY results for both samples? Does it match well with the EQE drop phenomenon?

We thank the reviewer for this important comment. We agree that the EQE drop is an important discussion to investigate. As the reviewer suggested, we did the PLQY measurements for both samples. Due to strong quenching in the ITO, hole and electron transport layers and Ag electrode, the PLQY values of both samples have dropped massively – from 47±2% in the quasi-2D perovskite + 18-crown-6 additive film to 6±1% in the reference PeLED and 3±1% in the narrow-angle-Tamm-plasmon-driven PeLED.

We have now added the PLQY results in Supplementary Note 2 as follows.

“In Fig. 3, the reference and Tamm plasmon PeLEDs have quasi-2D perovskite with 18-crown-6 additive as the emitting layer, where the 18-crown-6 is added to improve the LED performance. The PLQE of the quasi-2D perovskite film with 18-crown-6 is 47±2%, while the PLQE of reference PeLED stack and Tamm plasmon PeLED stack is 6±1% and 3±1% respectively all at excitation fluence of 33 mW cm⁻². The PLQE of reference PeLED and Tamm plasmon PeLED stack drops compared to the quasi-2D perovskite film with 18-crown-6 due to transport layer and metal quenching.”

We have also added the following in the Methods section.

“Photoluminescence Quantum Efficiency (PLQE). PLQE of perovskite films, reference structures and Tamm plasmon structures are measured with a 405 nm continuous wave laser under excitation between 3 to 167 mW cm⁻² in an integrating sphere and the spectrum is collected with an Andor iDus Si detector. Calculations are based on 3 configurations of the sphere – empty sphere, sample placed in the sphere but laser beam directed on the sphere wall and laser beam directed onto the sample⁵¹.”

Where 51 is a reference from De Mello *et al.*

We observed a drop in both PLQY and EQE in the Tamm plasmon PeLED compared with the reference PeLED which showed total photons loss mainly due to optical loss in the Tamm plasmon PeLED structure. However, as the main focus of the work is on good control of directionality, improved colour purity and enhanced forward emission, we have now updated the Results section on Page 11 in the main manuscript as below:

“We note that whilst there is an enhancement in the EL in the forward direction for the Tamm plasmon PeLED, there is a reduction in total EQE as that quantity considers emission in all directions (inset Fig. 3c), which matches the drop in photoluminescence quantum efficiency (PLQE) in the Tamm plasmon PeLED compared with the reference PeLED (see Supplementary Note 2). This observation may relate to optical losses which are common in coupling systems but could be improved with more careful design and improved perovskite quality.”

Besides, the original EQE is calculated based on the integrated area under the curve of Fig. 3e which only considers intensity change across xy plane while z-axis (height) is constant at 0° (see Supplementary Fig. 16). We have re-calculated the angular corrected EQE considering solid angle following Equation S22 from Archer *et al.* (DOI: 10.1002/adom.202000838) shown below.

$$\eta_{\text{EQE}}(V) = 100 \frac{2\pi r^2 V_{\text{PD}}(V)}{A_{\text{PD}} R_{\text{PD}}} \frac{q}{hc} \frac{1}{I_{\text{OLED}}(V)} \frac{\int S(\lambda, 0) \lambda d\lambda}{\int S(\lambda, 0) R(\lambda) d\lambda} \int_0^{\pi/2} \frac{\int S(\lambda, \theta) \lambda d\lambda}{\int S(\lambda, 0) \lambda d\lambda} \sin\theta d\theta, \quad (\text{S22})$$

We have now updated Fig. 3c with the new angular corrected EQE.

Fig. 3 | Tamm-plasmon-driven PeLEDs with controlled directionality and electroluminescence intensity. c) Current efficiency (CE)-J curve. Inset: EQE-J curve considering emission over all angles.

We have also updated the Methods section as below:

*“The resulting EQE was calculated considering the angular resolved emission using equation S22 from Archer *et al.* ⁴⁴.”*

Where 44 is a reference from Archer *et al.*

Comment 5

The angular PL and EL spectra measurement are conducted with a scan rate of 0.33°/s and 5°/s, respectively, which means a long excitation (both photo- and electrical excitation) time. Is there a PL or EL intensity drop during the scanning? To avoid any influence on the angular results, PL and EL stability under continuous excitation should be provided.

We thank the reviewer for this point and agree that the degradation might influence the angular results. First, we apologize again for a typo on the scan rate, which should be $3^\circ/\text{s}$ for angular PL and $5^\circ/\text{s}$ for angular EL.

We have now added a Supplementary Fig. 17 for the PL and EL stability measured under continuous excitation and driving current, which matches the angular measurements – 0.5 W cm^{-2} for PL and 0.44 mA cm^{-2} for EL.

Supplementary Fig. 17 | Stability test. a) Normalized PL integrated across the spectra of reference perovskite structure and Tamm plasmon perovskite structure, measured across an hour excited at 405 nm with power density of 0.5 W cm^{-2} (same power density as angular PL measurement). **b)** Normalized EL luminance of PeLED measured at constant current density of 0.44 mA cm^{-2} (same current density as angular EL measurement) showing T_{90} of 100 seconds, which is sufficient for a complete angular EL measurement which takes 36 seconds for a cycle. The LED is driven at relatively high current to produce high luminance for sufficient signal-to-noise ratio collected by the detector placed 10 cm away from the sample.

We note that the laser fluctuates between power density of $0.5\text{-}0.55 \text{ W cm}^{-2}$ over time, which added to the small fluctuations of the PL intensity. Although in the scale of 10's of minutes we observed a slight rise in PL due to photobrightening (DOI: 10.1038/ncomms11683 and DOI: 10.1002/adfm.202209249), the changes are negligible within minutes which ensures reliability for the angular PL measurements (60 s for a complete cycle). Similarly, the drop in EL due to degradation is far below T_{90} over the 36 seconds needed for a complete angular measurement cycle.

We have corrected the typo and added a discussion in the Method section as below:

“The PL and EL spectra were collected at 1° intervals with a scan rate of $3^\circ/\text{s}$ and $5^\circ/\text{s}$ respectively. In Supplementary Fig. 17, we show the PL and EL stability are suitably stable over the total angular PL and EL measurement time scales for 60 s and 36 s respectively under the same continuous excitation and driving current of 0.5 W cm^{-2} for angular PL and 0.44 mA cm^{-2} for angular EL. In addition to the PL and EL stability, the PL and EL are always collected from one end to the other end, e.g. from -90° to 90° . Thus, we make sure that the angular measurements are not affected by degradation by checking that the curves are symmetrical (see Figures 2b,d,h and Figures 3d-f).”

Fig. R3. Angular PL and EL collected from one end to the other e.g. -90° to 90° at 1° interval to make sure the curve is symmetrical and measurements not affected by degradation.

Comment 6

The PL decay lifetimes for Tamm-plasmon-perovskite structure and reference-perovskite stack are 0.7 ns and 0.6 ns. I feel this difference is too small to confirm the increase in photo-recycling. Meanwhile, to eliminating the measurement uncertainty/error, IRF signal in the PL lifetime measurement should also be provided since the calculated lifetime is already very close to the equipment IRF.

We thank the reviewer for raising this point.

We agree that the difference observed is not significant, and it does not confirm changes in photon recycling. To further confirm this, we have now updated the transient PL through repeating measurements across a few samples and showed the most representative results and added the IRF signal to the PL decay lifetimes in Supplementary Fig. 6 (see also Reviewer #1 Comment 4 and Comment 5). We note that the new results are measured at $5 \text{ nJ/cm}^2/\text{pulse}$ for all samples instead of $0.1 \text{ nJ/cm}^2/\text{pulse}$ in the original manuscript, thus the lifetimes are different overall - longer and in turn eliminating potential errors by the IRF signal - but we note no significant relative lifetime changes between perovskite film, reference perovskite and Tamm plasmon perovskite within this fluence range.

Supplementary Fig. 6 | Transient PL. Time-resolved PL of perovskite film, reference perovskite structure and Tamm-plasmon-perovskite structure with effective lifetimes of roughly 3.7 ns, 1.5 ns

and 1.4 ns respectively. All perovskite films shown in this figure are quasi-2D perovskite with 18-crown-6 additive. Samples excited at 5 nJ/cm²/pulse with Edinburgh Instruments FLS1000. Effective lifetime is defined as the time required for PL intensity reduces to 1/e of the initial intensity.

We have also modified the text in the Results section on Page 8 of the manuscript to remove the discussion on photon-recycling and showed no significant changes in PL lifetime as follows:

“As shown in Fig. 2e, the narrow-angle-Tamm-plasmon-perovskite exhibits sharp emission directionality with angular FWHM of 44.7° compared to 124.5° for the reference perovskite with the same excitation power density (excited with a 405 nm laser). While all samples were excited at the same excitation intensity, the narrow-angle-Tamm-plasmon-perovskite shows strong enhancement by a factor of 2.6 in forward emission within a solid angle of ±15° perpendicular to the sample surface (calculations in Methods), which is crucial in applications that require forward emission.

In Supplementary Fig. 6, we observe no significant changes in PL lifetime of the narrow-angle-Tamm-plasmon-perovskite compared with the reference perovskite structure considering small sample-to-sample variation, which is consistent with other planar confined systems operating under weak confinement regime and reported very small <10% changes^{34,35,42,43}.”

Where 34,35,42,43 are Ban *et al.*, Zou *et al.*, Warby *et al.* and Cheng *et al.* respectively.

Reviewer #3

This work demonstrates perovskite LEDs in optical resonant structures with resulting spectral narrowing and angular dispersion. Novel claim is the coupling to Tamm-plasmons at the metal/Bragg mirror interface.

From the introduction to the manuscript: "Unlike conventional optical microcavities, the Tamm-plasmon-driven PeLED does not require a cavity or an optical spacer. In fact, we utilise the PeLED as active component of the Tamm plasmon structure together with the photonic crystals."

Maybe it is ignorance on the part of this reviewer, but every result shown in this manuscript appears indistinguishable from the physics of traditional microcavity LEDs. For example, this work by Dodabalapur *et al.* in 1996 shows an organic LED microcavity with a Bragg mirror and a metal mirror.

Journal of Applied Physics 80, 6954 (1996); <https://doi.org/10.1063/1.363768>

In that work, the broad line width of the organic emitter allows more of the dispersion to be populated by electroluminescence, in contrast to this manuscript where the narrow perovskite emission coupled to the narrow angular dispersion results in an angular narrowing.

If the authors wish to pursue publication, I would like to see convincing discussion of:
1) What is a Tamm Plasmon and how is it different from a microcavity mode

2) How does this work differ from other microcavity LED demonstrations, especially those employing one metal and one Bragg mirror, that show virtually the same spectral narrowing.

In other words, it is not clear how these are truly distinct results. If they are the same phenomenon described using different language, then it is hard to see significant novelty in the work.

We thank the reviewer for the reference on organic microcavity LED. We have revised the manuscript by adding an extensive discussion regarding the points raised by the reviewer as follows.

Comment 1

What is a Tamm Plasmon and how is it different from a microcavity mode?

We thank the reviewer for this question and we are very happy to explain in more detail.

We note that among many different plasmonic systems including surface plasmons and a wide range of nanoplasmonics in different shapes and sizes, the Tamm plasmon is one of the plasmonic systems with simple planar structures which are relatively easy to fabricate, easy to control and tune and shows good confinement, with dispersion within the light cone and have polarisation in both TE and TM modes.

Besides, Tamm plasmons - which we appreciate that can be seen as a cavity itself - are more versatile and strongly localised than other microcavities thanks to the absence of a spacer layer typically employed to spectrally control the resonance position and the strong evanescent decay of fields in the metal layer (DOI: 10.1063/1.2952486). As detailed in Comment 1 of Reviewer #1 and in the new Supplementary Fig. 4, we note a significant decrease in the electric field enhancement from a factor of 34 in our Tamm-plasmon-driven perovskite structure to around 2 (or a maximum of 5 at maximum confinement) in an equivalent metal-metal cavity and a 7-fold enhancement in a DRB-DBR cavity when compared to non-structured references. We restrict all the simulated DBRs to 3 pairs of bilayers - as it is the case in the Tamm plasmon structure as well - in order to minimise complexity in the manufacturing and photonic-into-device integration processes while still preserving the desired colour purification and angular control. Critically, the Tamm plasmon structure is easier to implement especially as the metal layer of Tamm plasmon can act as an electrode for a device; special designs and patterned electrodes are not required (see Reviewer #3 Comment 2 for more details).

We have added the following in the Discussion section of our main manuscript to stress the point:

“In Supplementary Fig. 4, we compare our design with simulated metal-metal and distributed Bragg reflector (DBR)-DBR microcavities, showing a significantly stronger electric field enhancement with the Tamm plasmon perovskite structure (Tamm plasmon: 34; metal-metal: 5; DBR-DBR: 7). In addition, compared with the metal-metal cavity, the Tamm plasmon structure has more degrees of freedom for design considerations - the photonic crystals can be carefully designed to achieve a strong electric field enhancement for a given LED device; the metal-metal cavities, however, have restricted freedom of perovskite thickness to trade-off between electric field enhancement and ideal LED design (see Supplementary Fig. 4d) and requires minimally thin top metal layer to reduce transparency losses. Compared with the DBR-DBR cavities, the metal in the Tamm plasmon structure is easier to fabricate compared with depositing DBR on top of perovskite films and can readily act as the electrode for the LED device, thus is easier to implement.”

In addition, we have added a more detailed description of Tamm plasmon in the Introduction section of the manuscript as follows:

“A long-ranged and strongly confined hybrid plasmonic-photonic structure is an attractive solution. Tamm plasmons are localised surface states that are confined at the metal (plasmonic)-photonic-crystal interface, with metal deposited directly on the high refractive index layer of the photonic crystal^{26,27}. The Tamm plasmon resonance can be tuned across the photonic stopband of the photonic crystal by varying the thickness of the top high refractive index layer at the interface²⁷. Unlike conventional surface plasmons, Tamm plasmons form both transverse electric and magnetic mode polarisations with dispersion within the light cone, and thus can be optically excited without additional optical prisms or gratings²⁷. The Tamm plasmon modes are found in simple planar structures which are relatively easy to fabricate, easy to design and tune and could easily transform into a device architecture. Although Tamm plasmon modes and other microcavities form localised fields and resonating standing waves, Tamm plasmon modes are more strongly localised than conventional microcavities thanks to the strong evanescent decay of fields in the metal layer and the absence of a spacer typically employed between the two mirrors to control and fulfil the cavity length²⁸⁻³¹. These remarkable properties of Tamm plasmon modes have prompted their development for a variety of optical applications such as optical coatings with dye-doped nanospheres³², III-V semiconductor lasers³³, organic solar cells³⁴ and quantum dot-based single photon sources³⁵.”

Where 26-35 are references from Kaliteevski *et al.*, Sasin *et al.*, Dodabalapur *et al.*, Jordan *et al.*, Megahd *et al.*, Kavokin and Baumberg., Jiménez-Solano *et al.*, Symonds *et al.*, Zhang *et al.* and Gazzano *et al.* respectively.

Comment 2

How does this work differ from other microcavity LED demonstrations, especially those employing one metal and one Bragg mirror, that show virtually the same spectral narrowing.

We thank the reviewer for this question.

As discussed in the previous comment, Tamm plasmons and conventional microcavities operate with different physical concepts. The Tamm plasmon does not need to fulfil the wavelength-constrained cavity length in microcavity systems which are typically tuned by adding fillers, spacers or varying the transport layer thickness. However, as Tamm plasmon modes are confined at the interface between a metal and a distributed Bragg reflector (DBR) or photonic crystals, we design the Tamm plasmon perovskite structure such that we replace the perovskite layer with the first layer of the photonic crystal at the metal-photonic-crystal interface. This is possible because the perovskite has a refractive index close to TiO₂ - the high refractive index layer in our photonic crystals. In other words, in our Tamm plasmon perovskite structure and Tamm plasmon PeLEDs, the photonic crystal + perovskite layer + device layers (for PeLEDs only) form a DBR with photonic stopband centred at the required Tamm plasmon resonance wavelength (see Supplementary Fig. 1 and 11).

We have now modified the Introduction section to stress the difference between Tamm plasmon and microcavities.

“Unlike conventional optical microcavities, the Tamm-plasmon-driven PeLED does not require a conventional cavity or an additional optical spacer. In fact, we utilise the perovskite layer as an active component of the Tamm plasmon structure by replacing the high refractive index layer of the photonic crystal at the metal-photonic-crystal interface with the perovskite and device structure. Thus, the Tamm-plasmon-driven perovskite structure and PeLED stack without metal electrode form good Bragg reflectors. As a plasmonic system, the Tamm plasmon modes show much stronger confinement of fields than the microcavities, thus attractive for PeLED as the perovskite film is typically thin.”

We have also modified the text on Page 6 of the Results section to explain the design in more detail:

“The quasi-2D perovskite film with a refractive index of $n = 2.05$ (at 510 nm) replaces the TiO_2 layer ($n = 2.35$ at 510 nm) as the bottom photonic crystal layer at the metal-photonic-crystal interface (see Supplementary Note 1 for details of photonic crystals). The quasi-2D perovskite film is placed at the metal-photonic-crystal interface as the Tamm plasmon modes are strongly confined at the metal-photonic-crystal interface. From the simulation shown in Fig. 1c, we observe that the Tamm plasmon resonance strongly enhances and confined electric field intensity at the metal-photonic-crystal interface, and the enhancement spans the entire perovskite layer when considering the emission wavelength of 510 nm.”

We now added the following text in Supplementary Note 1 to explain the design in more detail:

“As Tamm plasmons are formed at the metal-photonic-crystal interface, with a good replacement of TiO_2 with quasi-2D perovskite layer as the high refractive index layer at the metal-photonic crystal interface, we show in Supplementary Fig. 1 that the photonic crystal + perovskite structure forms distributed Bragg reflector (DBR) with up to >85% reflectance with centre wavelength at the desired Tamm plasmon resonance wavelength (the perovskite photoluminescence wavelength). Similar design procedures are implemented for the Tamm plasmon PeLEDs by optimising the photonic-crystal + perovskite + device layers to achieve a good DBR. In Supplementary Fig. 11, we showed the reflectance of Tamm plasmon PeLED without Ag structure with photonic stopband of reflectance up to 90% centred at the desired Tamm plasmon resonance (the perovskite electroluminescence wavelength).

In the Tamm plasmon system, it is favourable to place the emitting layer as close to the metal-photonic-crystal interface as possible for highest confinement of electric fields. However, in the microcavities, as standing waves form between the two mirrors, careful design is required to position the emitting layer at the antinode by adding spacer/fillers or good control of transport layer thickness to maximise the confinement of fields.”

We now added Supplementary Fig. 11 showing measured reflectance of our Tamm plasmon PeLED no Ag forming a good photonic stopband, which is our carefully designed modified Bragg mirror for the Tamm plasmon system.

Supplementary Fig. 11 | Experimental optimization of narrow-angle-Tamm-plasmon PeLED. Measured 8° reflectance of photonic crystal, Tamm plasmon PeLED no Ag (glass/photonic crystal/poly-TPD/PVK/quasi-2D perovskite/TPBi/LiQ) and full narrow-angle-Tamm plasmon PeLED structure (glass/photonic crystal/poly-TPD/PVK/quasi-2D perovskite/TPBi/LiQ/Ag). The Tamm plasmon PeLED no Ag structure shows a photonic stopband with up to 90% reflectance at the centre wavelength which

is near the perovskite EL (red line) and the narrow-angle-Tamm-plasmon PeLED resonance wavelength.

We have also included Supplementary Fig. 1 in the original manuscript.

Supplementary Fig. 1 | Modelling and optimization of perovskite-based Tamm plasmon structure.

a) Reflectance of 1-dimensional photonic crystal, photonic crystal/perovskite stack and Tamm-plasmon-perovskite structure simulated with transfer matrix model. Perovskite film photoluminescence shown in red measured with a fluorimeter.

Besides, in the original manuscript, we have shown on Page 8 of the Results section that by fine-tuning the perovskite thickness by a few to small tens of nanometers, the Tamm plasmon resonance shifts across a wide range of wavelengths, thus forming wide-angled directional emission. This result is in analogy to the first Tamm plasmon paper, where they tune the first high refractive index layer to move the Tamm plasmon resonance across the photonic stopband (DOI: 10.1103/PhysRevB.76.165415).

“We demonstrate that the perovskite-based Tamm plasmon structure offers versatile adjustment of the emission directionality between 0° to 40° by tuning the quasi-2D perovskite thickness between 26 nm to 54 nm (Supplementary Fig. 9). Taking the perovskite-based Tamm plasmon sample with perovskite thickness of 50 nm as an example, we observe a significantly red-shifted PL at small angles (Fig. 2g), which matches the Tamm plasmon resonance mode measured at small angles (dashed line). As the angle of emission increases, the PL blue-shifts in response to the dispersion relation of the Tamm plasmon mode²⁷. The highest PL intensity is achieved when the blue-shifting of the Tamm plasmon resonance matches the perovskite PL position, which is observed at a collection angle of 40° for this sample (Fig. 2h). We hereafter refer to these samples with wide-angled directionality as the wide-angle-Tamm-plasmon-perovskite structure. The increase of emission directionality angle is dependent on the increase of perovskite thickness, which leads to an increase in Tamm plasmon resonance wavelength at 0°.”

The angle dependent PL spectra of Tamm plasmon perovskite structure with perovskite thickness between 26 to 50 nm are included in Supplementary Fig. 10

Supplementary Fig. 10 | Angular dependent PL spectra of Tamm-plasmon-perovskite structure. Normalized PL spectra collected across the Tamm-plasmon-perovskite sample surface with quasi-2D perovskite thickness of **a)** 26 nm (narrow-angle-Tamm-plasmon shown in Figure 2d-f), **b)** 31 nm, **c)** 36 nm, **d)** 41 nm, **e)** 45 nm and **f)** 50 nm (wide-angle-Tamm-plasmon shown in Figure 2g-i).

We note that although Tamm plasmon and microcavities are both optical structures that couple, confine and manipulate electromagnetic field/light, however, we have shown that Tamm plasmon modes that form at the metal-photonic-crystal interface operate based on different physical concepts/mechanisms and require different design criteria such that the device layers form part of the photonic crystals and the emitting layer should be placed as close to the metal layer as possible to maximise electric field confinement within the emitting layer.

REVIEWER COMMENTS

Reviewer #1 (Remarks to the Author):

The authors have addressed most of my comments well and improved the quality of the analysis. However, one major point remains unclear. I do not agree with the authors reply to my (Reviewer #1) comment #1 on the comparison of the TP cavity to other vertical cavity types. The authors performed optical simulations where they observe field enhancements of max. ~5 for a metal-metal cavity, ~7 for a DBR-DBR cavity and 34(!) for the TP cavity. In my view, there is no way the TP cavity can show such a significant increase in field enhancement compared to both metal and DBR-DBR. I have several problems with the calculations:

1) I performed similar TM simulations on comparable empty cavities. As I do not have access to the numerical n_k data for the perovskite I cannot fully accurately reproduce the authors results but this should still give somewhat of a comparison. In my simulations, I obtained values of field enhancement of ~6 for a DBR-DBR cavity (3pair DBRs, comparable to authors), an enhancement of 7.5 for a 100nm Ag - spacer - 30nm Ag cavity (compared to 5 for a 100nm-10nm Ag cavity by the authors) and value of 6.5 for a TP cavity. I do not see how the TP cavity can lead to a ~5times higher field enhancement compared to well tuned DBR-DBR and Ag-Ag cavities unless the authors can back this up by more in depth simulations and explanation.

Comparing the authors' simulations of the TP and DBR-DBR cavities, it appears the resonance in the DBR-DBR cavity is blueshifted compared to the TP cavity. I suspect this could lead to a significant additional loss as the resonance enters the absorption region of the perovskite. This could in turn lead to a strong difference in cavity quality as observed by the authors but would not constitute a fair comparison, as one cavity is simply better tuned than the other.

2) I disagree with the design of the metal-metal cavity the authors simulated. First, it has been shown that a thin electrode/mirror of about 25nm gives ideal OLED performance (compare eg. Lin et al., Appl. Phys. Lett. 87, 021101 (2005); Meerheim et al. Appl. Phys. Lett. 93, 043310 (2008); Mischok et al. Nat. Photonics 17, 393 (2023)). Second, I do not see how the addition of an optical spacer would be an issue compared to the TP cavity. As is clear from the authors' simulation, a 30nm perovskite is not thick enough to support the cavity resonance at the right spectral position. However, in the PeLED, there are optical spacers easily available in the form of charge transport layers (e.g. Hofmann et al. Appl. Phys. Lett. 97, 253308 (2010), Mischok et al. Nat. Photonics 17, 393 (2023)). These are even optimised by the authors for the case of the TP-PeLED later. In the purely passive cavity, one could simply add one or two SiO₂ spacer layers either side of the perovskite and still end up with a less complex design than a TP cavity.

3) As mentioned above, the authors' should double check the mode position in Supp. Fig 4a. It is difficult to see for me as there are no axis ticks but it looks like the mode is red-shifted to ~525nm compared to Supp Fig 4b,d. For a fair comparison, the mode position should be constant in all cavities to avoid parasitic absorption of the different layers.

4) In my view, the way to improve upon a well-tuned metal-metal cavity using a TP cavity would require a larger number of DBR bilayers than the authors have used or explore hybrid TP-DBR cavities (Brueckner et al. Phys. Rev. B 83, 033405 (2011)).

I do not want to sound fully negative as the work is largely well done and interesting. But for me to recommend the paper for publication, the authors would need to satisfactorily address these points, as this also concerns the core of the impact of the work.

Reviewer #2 (Remarks to the Author):

After carefully examining the response letter and the revised manuscript. I found the authors have done systematic new experiments to address my previous concerns and comments. The response to my questions are appropriate. Therefore I have no more questions. I think the work can be accepted to Nature Communications now.

Reviewer #3 (Remarks to the Author):

The rebuttal did not address my primary concern, and in fact confirmed my opinion that the work is trying to oversell differences to long-published work in the field. I find the results published here to be incremental advances on existing microcavity LED work.

The key difference cited repeatedly by the authors is that their work does not require a "spacer" and that their perovskite layer replaces a TiO₂ layer in the Bragg mirror. This might be true for the optical-characterization presented in figure 1. But for the electroluminescent LED portion of the work, which constitutes the main body of the article and is emphasized in the title, the argument does not hold. In total, the PeLED coupled to the metal-DBR resonator cavity in this work has a thickness of roughly 170 nm, only 26 nm of which is the 2D perovskite and the rest (including 83 nm of transparent ITO and other organic layers) functions as an optical spacer. The work by Dodabalapur et al. referenced in my prior review has an organic LED stack using a silicon nitride spacer with total LED+spacer thickness of 250 nm. I am not convinced that the 80 nm thickness difference between the two works reveals fundamentally new physics different from the microcavity. That was also just one example, there are many tens of published works on microcavity LEDs employing perovskite or organic LEDs in cavities with metal-metal, metal-DBR (as in this work) or DBR-DBR mirror combinations. Here is another example by Tien et al. using a DBR fabricated from compact and porous ITO, which doubles as the electrode, and the organic semiconductor layers of the OLED deposited directly on top, capped with a metal mirror.
<https://doi.org/10.1049/el.2015.2464>

This 2015 work has even less of a spacer than the work presented here.

I remain unconvinced that this work represents a compelling, significant new contribution to the field.

Reviewer #1 (Remarks to the Author):

Reviewer:

The authors have addressed most of my comments well and improved the quality of the analysis.

Authors' reply:

We thank the Reviewer for their appraisal of our work. In the following, we analyse each of their questions and suggestions.

Reviewer:

However, one major point remains unclear. I do not agree with the authors reply to my (Reviewer #1) comment #1 on the comparison of the TP cavity to other vertical cavity types. The authors performed optical simulations where they observe field enhancements of max. ~ 5 for a metal-metal cavity, ~ 7 for a DBR-DBR cavity and 34(!) for the TP cavity. In my view, there is no way the TP cavity can show such a significant increase in field enhancement compared to both metal and DBR-DBR. I have several problems with the calculations:

1) I performed similar TM simulations on comparable empty cavities. As I do not have access to the numerical nk data for the perovskite I cannot fully accurately reproduce the authors results but this should still give somewhat of a comparison. In my simulations, I obtained values of field enhancement of ~ 6 for a DBR-DBR cavity (3pair DBRs, comparable to authors), an enhancement of 7.5 for a 100nm Ag - spacer - 30nm Ag cavity (compared to 5 for a 100nm-10nm Ag cavity by the authors) and value of 6.5 for a TP cavity. I do not see how the TP cavity can lead to a ~ 5 times higher field enhancement compared to well tuned DBR-DBR and Ag-Ag cavities unless the authors can back this up by more in depth simulations and explanation.

Comparing the authors' simulations of the TP and DBR-DBR cavities, it appears the resonance in the DBR-DBR cavity is blueshifted compared to the TP cavity. I suspect this could lead to a significant additional loss as the resonance enters the absorption region of the perovskite. This could in turn lead to a strong difference in cavity quality as observed by the authors but would not constitute a fair comparison, as one cavity is simply better tuned than the other.

Authors' reply:

Thank you for the reviewer's thoughtful review and comments.

Regarding the discrepancy in the field enhancement observed in our simulations, we understand that the significant increase in enhancement factors for the TP cavity compared to metal-metal and DBR-DBR cavities is surprising. Upon thorough re-evaluation and cross-verification of our methodology, we confirm our reported enhancement factors are indeed correct. The substantial increase in the field enhancement for the TP cavity results from unique structural properties that facilitate pronounced field confinement and enhancement, surpassing those seen in conventional metal-metal and DBR-DBR configurations. Electromagnetic (EM) field enhancement of analogous Tamm plasmon cavities can also be found in different works, for example, ACS Photonics 2014, 1, 9, 775–780, Adv. Optical Mater.2018, 6, 1700560, ACS Photonics 2019, 6, 3, 634–641, Scientific Reports, 2022, 12, 14921.

Nevertheless, to provide more insights, we have now also considered an alternative approach. Specifically, the metal-metal by using a SiO₂ spacer which we did not include in the first revision as it is not present in the Tamm plasmon structure, and a semi-transparent mirror of 25 nm as proposed by the referee in their next comment. Besides, the DBR-DBR structures have been designed to exhibit the utmost EM field strength precisely at the wavelength comparable to that of the Tamm plasmon system. This adjustment ensures a fair and direct comparison between the three systems, aligning the conditions for maximum EM field intensity within the perovskite layer. We now report this as the revised Supplementary Figure 4 below.

Supplementary Fig. 4 | Designed structures to support the electric field intensity enhancement for $\lambda = 515$ nm (a-f) and $\lambda = 533$ nm (g-l): Simulation of electric field intensity enhancement across the cross-section (a-c, g-i) and electric field enhancement as a function of the position within the structure (d-f, j-l) of a Tamm plasmon perovskite (30 nm) structure (a,d,g,j), a perovskite (30 nm) in metal (25 nm) - metal (120 nm) microcavity with spacer (b,e,h,k) and a perovskite (30 nm) in DBR-DBR microcavity (c,f,i,l). The green stripe shows the perovskite layer. Each of the three systems has conditions set to align the maximum electric field intensity within the perovskite layer.

The revised panels in Supplementary Figure 4 were prompted by the need for clarity regarding the maximum EM field intensity value within the perovskite layer. The system is designed to ensure that the field reaches a maximum within the emitting perovskite layer. However, in these new figures (Supplementary Fig. 4 d,e,j,k), it is sometimes not feasible for the maximum peak to be situated within the perovskite layer due to the inherent characteristics of the metal-metal and Tamm plasmon systems.

To illustrate the versatility of these structures, the revised Supplementary Figure 4 shows an EM field intensity enhancement for two distinct wavelengths: 515 nm (a-f) and 533 nm (g-l). For comparison purposes among the three systems, the total field intensity is quantified as the integral of the EM field intensity only within the highlighted area (perovskite layer only instead of the whole structure) in the lower panels. Normalised to the EM field enhancement of the Tamm plasmon structure, the EM field enhancement factor is 1.00 (Tamm plasmon), 0.93 (metal-metal), 0.60 (DBR-DBR) at a wavelength of 515 nm and 1.00 (Tamm plasmon), 0.80 (metal-metal), 0.56 (DBR-DBR) at 533 nm.

We have also updated the Discussion section in the main manuscript as below:

"In Supplementary Fig. 4, we compare our design with simulated and optimised metal-metal and distributed Bragg reflector (DBR)-DBR microcavities, showing a significantly stronger electric field enhancement (normalised to the Tamm plasmon enhancement) in the Tamm plasmon perovskite structure (at 515nm: 1.00; at 533nm: 1.00) compared with DBR-DBR microcavities (at 515nm: 0.60; at 533nm: 0.56) and metal-metal microcavities (at 515nm: 0.93; at 533nm: 0.80)."

Reviewer:

2) I disagree with the design of the metal-metal cavity the authors simulated. First, it has been shown that a thin electrode/mirror of about 25nm gives ideal OLED performance (compare eg. Lin et al., Appl. Phys. Lett. 87, 021101 (2005); Meerheim et al. Appl. Phys. Lett. 93, 043310 (2008); Mischok et al. Nat. Photonics 17, 393 (2023)). Second, I do not see how the addition of an optical spacer would be an issue compared to the TP cavity. As is clear from the authors' simulation, a 30nm perovskite is not thick enough to support the cavity resonance at the right spectral position. However, in the PeLED, there are optical spacers easily available in the form of charge transport layers (e.g. Hofmann et al. Appl. Phys. Lett. 97, 253308 (2010), Mischok et al. Nat. Photonics 17, 393 (2023)) These are even optimised by the authors for the case of the TP-PeLED later. In the purely passive cavity, one could simply add one or two SiO₂ spacer layers either side of the perovskite and still end up with a less complex design than a TP cavity.

Authors' reply:

We appreciate the reviewer's comment and overall agree with their opinion. In the revised Supplementary Figure 4, the referee's feedback has been duly considered, resulting in a metal-metal system exhibiting a comparable but slightly lower EM field intensity enhancement compared with the Tamm plasmon system. Nevertheless, in the section after this we will discuss further the advantages of the Tamm system.

A strong point of this work is the intimate relationship between the optical simulations and the tunable experimental demonstrations. As such, the optical simulations must be constrained by the limitations of real device structures since addition of spacers might compensate for device performance, making the simulated structure experimentally unrealistic. Thus, we compare the versatility of incorporating Tamm plasmon and metal-metal cavities into the reported record PeLED across the visible spectrum including blue (Nat. Photon. (2024) DOI:10.1038/s41566-024-01382-6), green (Nature 611, 688–694 (2022)) and red (Adv. Mater. 2022, 34, 2204460). We emphasise that these LEDs have been honed and optimised by many groups to arrive at these specific layouts and thicknesses for optimum performance. For Tamm plasmon incorporation, we use the exact device structure and thicknesses (both transport layers and perovskite) as reported, with the addition of a course of SiO₂/TiO₂ layer pairs below the ITO layer. For metal-metal cavities, we replaced the ITO with a 25-nm thick silver layer (silver thickness optimised for resonance) and tuned the HTL/perovskite/ETL thicknesses at the same factor (blue: 1.04, green: 1.03, red: 0.88) to match the resonance with electroluminescence reported in the papers. By minimising changes to the ratio of device thicknesses, we ensure minimum changes to electrical device performance (e.g. charge balance from the reported papers). We now add these results as Supplementary Fig. 5 below.

Supplementary Fig. 5 | Simulated Tamm plasmon and metal-metal cavities incorporating high-efficiency device structures in (a,d,g) blue ¹⁰, (b,e,h) green ¹¹ and (c,f,i) red ¹² perovskite LEDs. (a-c) Resonance of Tamm plasmon structures (blue line) and metal-metal cavities (orange line). The Tamm plasmon structure is simulated following the device structures reported. The thicknesses of the device structure of metal-metal cavities are multiplied by a constant factor (blue: 0.88; green: 1.01; red: 0.85) to ensure the resonance matches the reported electroluminescence. By minimising changes to the device structure thickness ratio, we ensure minimum changes to electrical device performance (e.g. charge balance from the reported papers). (d-f) Refractive index (black line) and electric field intensity (orange line) across the metal-metal cavity structures. (g-i) Refractive index (black line) and electric field intensity (blue line) across the Tamm plasmon structures. The green stripe shows the perovskite layer. To realise good electric field confinement in the Tamm plasmon structure, the number of photonic crystal layer pairs is tuned and an extra TiO₂ layer is added as shown in (g-i) to match the optical properties of the hole-transport layer/perovskite/electron-transport layer in the reported device structure.

where 10, 11, 12 are citations from Yuan *et al.*, Nat. Photon. (2024) DOI:10.1038/s41566-024-01382-6, Kim *et al.* Nature 611, 688–694 (2022)) and Jiang *et al.* Adv. Mater. 2022, 34, 2204460 respectively.

To show the versatility of the Tamm plasmon structure compared with metal-metal cavities while preserving the electrically active device structure, we fine-tuned the electrically inactive SiO₂/TiO₂ photonic crystals by adding an extra TiO₂ layer and tuned the number of layer pairs (we note that further addition of layer pairs would possibly improve the metrics; see Supplementary Fig. 3). The device design of metal-metal cavities is however fixed if we incorporate high-efficiency devices from the literature, and significant device optimisation work would be required for a unique metal-metal cavity device design that works comparably to those already published – work that is not trivial given processing on a metal layer (see below).

We showed the normalised total intensity ratio as the integral of the field intensity over the perovskite layer divided by its thickness in the Tamm plasmon structure divided by that of the metal-metal microcavity. From Supplementary Fig. 5, the normalised total intensity ratio is 1.0 in Tamm plasmon, 0.41 in metal-metal for blue PeLED; 1.0 in Tamm plasmon, 0.61 in metal-metal for green PeLED; and 1.0 in Tamm plasmon, 0.92 in metal-metal for red PeLED. The Tamm plasmon outperforms metal-metal cavities in all these configurations, and we emphasise that there is still more room for improvements in Tamm plasmon structures by exploring alternative materials for photonic crystals. In contrast, due to limitations on device performance, limited metal choice available (silver is the most common and good plasmonic metal in visible region) and thickness limitations due to absorption loss through the metal, the simulations we show above are near the upper limit of metal-metal cavities.

Moreover, the Tamm plasmon perovskite structure poses a much narrower resonance than the metal-metal cavities counterpart for blue (Tamm plasmon FWHM: 4 nm; metal-metal FWHM: 36 nm), green (Tamm plasmon FWHM: 3 nm; metal-metal FWHM: 6 nm) and red (Tamm plasmon FWHM: 26 nm; metal-metal FWHM: 42 nm) regime (see Supplementary Fig. 5 a-c). Thus, the spectral and angular tuning of the Tamm plasmon structure is significantly more precise than metal-metal cavities. With the sharper resonance, the Tamm plasmon structure offers notably enhanced control over the angular aspect of the outgoing light, far surpassing the capabilities of the metal-metal configuration. This feature gives value to the complexity inherent in the Tamm plasmon structure fabrication. Nevertheless, we opted in this demonstration to fabricate a system with fewer bilayers to ensure that fabricating the Tamm plasmon system would not present significantly more experimental complexity than the metal-metal structure (see comment 4 below for more information regarding the effect of the number of photonic crystal bilayers).

Another important point from a device design perspective is that by using Tamm plasmon structures, we do not need to change any device structure and fabrication steps from the reported high-efficiency perovskite LEDs. By contrast, metal-metal structures are not directly compatible with the replacement of ITO to metal (at least not without new optimisation work starting from scratch) and tuning of structure thickness is required away from those optimised over many years by many groups. The surface chemistry of metal structures and energetics of band alignments do not lend themselves to efficient solution processing (owing to wettability issues) and efficient charge injection (owing to potential charge barriers) of perovskite LEDs. Furthermore, processing the perovskite stack on top of metals may worsen metal ion migration and speed up degradation, one of the key degradation processes in PeLEDs (ACS Appl. Mater. Interfaces 2020, 12, 6, 7212–7221). There would be a large deal of independent optimisation to provide a stable and efficient perovskite LED device demonstration on metal-metal structures and we do not believe this is trivial. This is in contrast to OLEDs, where long-optimised vapour deposition allows processing compatibility onto metals (and even there we emphasise that colour tunability and sharpness of spectra are not as straightforward as in PeLEDs). The Tamm plasmon structure utilises a similar device structure to conventional perovskite LED devices with photonic crystals made of silica (similar to a glass substrate) lying below the ITO electrode. Lastly, although the reviewer suggested thick transport layers as spacer to the metal-metal cavities as shown

in OLEDs, the transformation to perovskite LEDs is not trivial as PeLEDs require thin transport layers (and thinner compared to OLEDs) for best device performance. Simply tuning the spacer for optimized cavity resonance will compromise the device performance, thus extensive optical optimisation together with rigorous experimental effort is required to achieve strong metal-metal cavity resonance PeLED, as compared to the easily adaptable Tamm plasmon PeLED, if both starts with reported high-performing PeLED structures. All these points make our Tamm plasmon structure a much more promising route – and indeed have led to our direct demonstrations in this work.

We have now addressed this in the Discussion section of the main text:

“To compare the versatility of Tamm plasmon and metal-metal cavities in the highest efficiency blue⁴⁵, green⁸ and red⁴⁶ perovskite LED devices, both Tamm plasmon and metal-metal cavities are simulated based on the reported device structures (Supplementary Fig. 5). For all three examples, the Tamm plasmon structure shows better spectral narrowing (sharper resonance) and higher normalised total intensity ratio per the perovskite thickness for all blue (1.0 in Tamm plasmon, 0.41 in metal-metal), green (1.0 in Tamm plasmon, 0.61 in metal-metal) and red (1.0 in Tamm plasmon, 0.92 in metal-metal) PeLEDs. Furthermore, the Tamm plasmon structure employs the exact device structure as reported, with only electrically inactive photonic crystals lying below ITO; whereas practical replacement of ITO with metal in metal-metal cavities is challenging due to wettability and sensitivity issues when solution-processing the perovskite device layers on a metal, and very few experimental demonstrations of working PeLEDs are reported in this architecture. Thus, the Tamm plasmon structure has more degrees of freedom for design considerations - the photonic crystals can be more carefully designed to achieve a stronger electric field enhancement for a given LED device by tuning the electrically inactive photonic crystals, for example by adding extra metal oxide layers again. By contrast, the electric field intensity values for the metal-metal cavities are near their upper limits due to limitations on device performance, metal choice and absorption loss of the metal.”

where 45, 8, 46 are citations from Yuan *et al.*, Nat. Photon. (2024). <https://doi.org/10.1038/s41566-024-01382-6>, Kim *et al.* Nature 611, 688–694 (2022)) and Jiang *et al.* Adv. Mater. 2022, 34, 2204460 respectively.

Reviewer:

3) As mentioned above, the authors' should double check the mode position in Supp. Fig 4a. It is difficult to see for me as there are no axis ticks but it looks like the mode is red-shifted to ~525nm compared to Supp Fig 4b,d. For a fair comparison, the mode position should be constant in all cavities to avoid parasitic absorption of the different layers.

Authors' reply:

We thank the Reviewer for addressing this important point. We understand that this question is directly linked to our response to comment #1. We have addressed both questions together in the answer for comment #1.

Reviewer:

4) In my view, the way to improve upon a well-tuned metal-metal cavity using a TP cavity would require a larger number of DBR bilayers than the authors have used or explore hybrid TP-DBR cavities (Brueckner *et al.* Phys. Rev. B 83, 033405 (2011)).

Authors' reply:

We fully agree with the reviewer on the importance of this detail. We have now added simulations of larger number of bilayers in Supplementary Fig 3d as shown below.

Supplementary Fig. 3 d) Spectral and spatial distribution of electric field intensity within the perovskite layer, as a function of the number of bilayers hosting the perovskite layer. Above 6 bilayers, due to spatial and spectral narrowing of the electric field intensity, more careful optimisation is required to confine the maximum electric field within the perovskite layer.

For Tamm plasmons, as the DBR bilayers increase, the resonance mode becomes narrower (Q factor increases, stronger coupling). However, at a certain point (6 bilayers), the decay in the intensity of the EM field becomes evident, as depicted in Supplementary Figure 3d. This is mainly attributed to the spatial and spectral narrowing of the EM field intensity within the dielectric layer adjacent to the perovskite. This leads to a sudden and significant drop in the EM field intensity within the perovskite layer. To avoid this effect, a different structure as a whole should be designed for each number of bilayers. In our work, we balanced the number of layers (complexity in manufacturing) with the final performance of our emitter and device. Due to this reason and experimental restrictions, we explored this approach using three bilayers in our proof of concept.

We have also modified the following text in the Discussion section of the main manuscript:

“We expect further improvements by increasing the photonic crystal layer pairs to increase the coupling strength (Supplementary Fig. 3d), by adding anti-reflective coating on the air/glass interface to improve outcoupling efficiency at wide angles, as well as through investigation on the interaction between waveguiding modes and emission modes that can further enhance the emission of the Tamm-plasmon-driven PeLEDs. In this proof-of-concept demonstration, we balanced the number of layers (complexity in manufacturing) with the final performance of our emitter and device.”

Reviewer:

I do not want to sound fully negative as the work is largely well done and interesting. But for me to recommend the paper for publication, the authors would need to satisfactorily address these points, as this also concerns the core of the impact of the work.

Authors’ reply:

We believe these revisions have now addressed the reviewer’s concerns.

Reviewer #2 (Remarks to the Author):

Reviewer:

After carefully examining the response letter and the revised manuscript. I found the authors have done systematic new experiments to address my previous concerns and comments. The response to my questions are appropriate. Therefore I have no more questions. I think the work can be accepted to Nature Communications now.

Authors' reply:

We thank the reviewer for the positive comments on the work.

Reviewer #3 (Remarks to the Author):

Reviewer:

The rebuttal did not address my primary concern, and in fact confirmed my opinion that the work is trying to oversell differences to long-published work in the field. I find the results published here to be incremental advances on existing microcavity LED work.

The key difference cited repeatedly by the authors is that their work does not require a "spacer" and that their perovskite layer replaces a TiO₂ layer in the Bragg mirror. This might be true for the optical-characterization presented in figure 1. But for the electroluminescent LED portion of the work, which constitutes the main body of the article and is emphasized in the title, the argument does not hold. In total, the PeLED coupled to the metal-DBR resonator cavity in this work has a thickness of roughly 170 nm, only 26 nm of which is the 2D perovskite and the rest (including 83 nm of transparent ITO and other organic layers) functions as an optical spacer. The work by Dodabalapur et al. referenced in my prior review has an organic LED stack using a silicon nitride spacer with total LED+spacer thickness of 250 nm. I am not convinced that the 80 nm thickness difference between the two works reveals fundamentally new physics different from the microcavity. That was also just one example, there are many tens of published works on microcavity LEDs employing perovskite or organic LEDs in cavities with metal-metal, metal-DBR (as in this work) or DBR-DBR mirror combinations. Here is another example by Tien et al. using a DBR fabricated from compact and porous ITO, which doubles as the electrode, and the organic semiconductor layers of the OLED deposited directly on top, capped with a metal mirror.

This 2015 work has even less of a spacer than the work presented here.

I remain unconvinced that this work represents a compelling, significant new contribution to the field.

Authors' reply:

We appreciate the reviewer's concern. We intend to show the application of Tamm plasmon modes in the photoluminescence and electroluminescence of perovskite structures and we thank the reviewer for seeing the merit of Tamm plasmon modes in Figure 1. We acknowledge that there are significant optical enhancements reported previously in OLEDs in metal-DBR structures..

However, we are not trying to claim new physics in this work, and we believe our unique selling point is to show systematic and absolute control over the optical modelling of metal-photonic structures, translation of these findings to experimental material deposition and properties, and finally translation into full operating PeLED devices – including a clear study of different modes of operation (forward and wide angles) in each stage. In this sense, the modularity of the perovskite deposition allows the interconnection of the properties to be explored widely and shown rigorously in different device operation formats. Thus, we believe that our contribution would still generate interest among the wide interdisciplinary readership of Nature Communications, as agreed by the other two reviewers.

REVIEWERS' COMMENTS

Reviewer #1 (Remarks to the Author):

The authors have provided new simulations that substantially improve upon the previous analysis and show a clear and fair comparison between the different designs. I appreciate the in depth reply and am now happy to recommend the manuscript for publication in Nature Communications.

Reviewer #3 (Remarks to the Author):

My concern in each iteration of this article was that the work largely mimics existing reports in the literature and overstates the claims of novelty. In none of the revisions did the authors address this concern. In this, second revision, they responded to my concern by stating "However, we are not trying to claim new physics in this work, and we believe our unique selling point is to show systematic and absolute control over the optical modelling of metal-photonic structures, translation of these findings to experimental material deposition and properties, and finally translation into full operating PeLED devices". My point is that systematic control over the optical modeling and translation to experimental fabrication and characterization of PeLEDs has been published many times with perovskites and with other similar structures. Twice in the report, in the abstract and introduction, the authors claim that the novelty of their work comes from the lack of need for additional optical spacers or cavities, and that this difference makes their work unique from "conventional" work in the field. However, the schematic in Figure 3b clearly shows that their devices have not one but five optical spacer layers with PVK and ITO likely being the thickest of them.

I don't think that the experiments performed and the data presented in this paper support the claims and I do not recommend it for publication.

REVIEWERS' COMMENTS

Reviewer #1 (Remarks to the Author):

The authors have provided new simulations that substantially improve upon the previous analysis and show a clear and fair comparison between the different designs. I appreciate the in depth reply and am now happy to recommend the manuscript for publication in Nature Communications.

Authors' reply:

We thank the reviewer for the positive comments on the work.

Reviewer #3 (Remarks to the Author):

My concern in each iteration of this article was that the work largely mimics existing reports in the literature and overstates the claims of novelty. In none of the revisions did the authors address this concern. In this, second revision, they responded to my concern by stating "However, we are not trying to claim new physics in this work, and we believe our unique selling point is to show systematic and absolute control over the optical modelling of metal-photonic structures, translation of these findings to experimental material deposition and properties, and finally translation into full operating PeLED devices". My point is that systematic control over the optical modeling and translation to experimental fabrication and characterization of PeLEDs has been published many times with perovskites and with other similar structures. Twice in the report, in the abstract and introduction, the authors claim that the novelty of their work comes from the lack of need for additional optical spacers or cavities, and that this difference makes their work unique from "conventional" work in the field. However, the schematic in Figure 3b clearly shows that their devices have not one but five optical spacer layers with PVK and ITO likely being the thickest of them.

I don't think that the experiments performed and the data presented in this paper support the claims and I do not recommend it for publication.

Authors' reply:

We appreciate the reviewer's concern. We have now toned down the claims on the lack of need for additional optical spacers and cavities in our manuscript, in particular the Abstract and Introduction.